# Cell type diversity in a developing octopus brain

Ruth Styfhals [1,2], Grygoriy Zolotarov[3], Gert Hulselmans[4,5], Katina I. Spanier [4,5], Suresh Poovathingal[5], Ali M. Elagoz [1], Seppe De Winter [4,5], Astrid Deryckere [1,7], Nikolaus Rajewsky [3,6], Giovanna Ponte [2], Graziano Fiorito [2], Stein Aerts [4,5] & Eve Seuntjens [1] ✉

Octopuses are mollusks that have evolved intricate neural systems comparable with vertebrates in terms of cell number, complexity and size. The brain cell types that control their sophisticated behavioral repertoire are still unknown. Here, we profile the cell diversity of the paralarval *Octopus vulgaris* brain to build a cell type atlas that comprises mostly neural cells, but also multiple glial subtypes, endothelial cells and fibroblasts. We spatially map cell types to the vertical, subesophageal and optic lobes. Investigation of cell type conservation reveals a shared gene signature between glial cells of mouse, fly and octopus. Genes related to learning and memory are enriched in vertical lobe cells, which show molecular similarities with Kenyon cells in *Drosophila*. We construct a cell type taxonomy revealing transcriptionally related cell types, which tend to appear in the same brain region. Together, our data sheds light on cell type diversity and evolution in the octopus brain.

Cephalopods, such as cuttlefish, squid, and octopus, are enigmatic organisms that have evolved impressive cognitive capabilities. They can display a range of complex behaviors like problem-solving, tool use, and millisecond camouflaging skills, for which higher cognitive functions are likely required[1–4]. Although the basic design of an octopus brain is typically molluscan, with a neuropil surrounded by a layer of monopolar neuronal cell bodies, its anatomical complexity is unparalleled among invertebrates[5]. Octopuses have a large centralized brain with more than 30 differentiated lobes and an intricate organization to support the transfer, integration, and computation of information[6,7].

The octopus brain consists of (1) two optic lobes, involved in visual sensory processing and memory storage of visual information; (2) the supraesophageal mass (sem), a sensory-motor, associative and integrative center, which contributes to long-term memory storage; and (3) the subesophageal mass (sub), responsible for motor and visceral coordination and other sensory processing[6]. The central nervous system of an adult octopus counts about 200 million cells, which is comparable to the number of neurons in the brain of a tree shrew[8,9]. The optic lobe (ol) consists of a cortex and a medulla. Its laminated cortex, also referred to as 'deep-retina', contains different types of amacrine and bipolar cells, while medulla neurons are organized in islands of unipolar nerve cells, and small and large tangential cells[6,10]. Within the central sem and sub masses, there is heterogeneity of cell sizes between different regions, but also within regions. The vertical lobe, which has been posited to be the functional analog of the invertebrate mushroom body and the mammalian pallium[5,11], contains 26 million neurons that are mostly small amacrine cells but also a few large projecting neurons close to the neuropil[6,9].

Although cell types present in the adult octopus brain have been characterized morphologically by J.Z. Young[6,10], the precise number of functionally different cell types is unknown, and their molecular

[1]Laboratory of Developmental Neurobiology, Department of Biology, KU Leuven, Leuven, Belgium. [2]Department of Biology and Evolution of Marine Organisms, Stazione Zoologica Anton Dohrn, Naples, Italy. [3]Laboratory for Systems Biology of Gene Regulatory Elements, Berlin Institute for Systems Biology, Max Delbrück Center for Molecular Medicine in the Helmholtz Association, Hannoversche Str. 28, 10115 Berlin, Germany. [4]Department of Human Genetics, KU Leuven, Leuven 3000, Belgium. [5]VIB Center for Brain & Disease Research, KU Leuven, Leuven 3000, Belgium. [6]Department of Pediatric Oncology/Hematology, Charité-Universitätsmedizin Berlin, Berlin, Germany. [7]Present address: Department of Biological Sciences, Columbia University, New York, US. ✉e-mail: eve.seuntjens@kuleuven.be

signature is unresolved. Data from several studies suggest that dopamine[12], GABA[13], serotonin[14], peptides (FMRFa[15,16], VD1/RDP2[17]), and nitric oxide[18] may act as neurotransmitters and/or neuromodulators during cephalopod development, suggesting the existence of a large variety of cell types.

The advent of genomic data now makes a deeper analysis of cell-type diversity possible. The first octopus genome has highlighted the expansion of gene families such as protocadherins (PCDH), G-protein coupled receptors (GPCR), and Zinc-finger transcription factors (ZnF) in *Octopus bimaculoides*[19]. These gene families have known roles in brain development and neural wiring in complex-brain species[20–22]. Whether these peculiar expansions have contributed to cell-type diversity in octopus is unknown.

The common octopus, *Octopus vulgaris*, lays hundred of thousands of transparent eggs, which take ~1 month to complete embryonic development and hatching[23]. At this point, free swimming paralarvae undergo a planktonic phase before they adopt the benthic lifestyle[24]. Their brain develops from an embryonic neurogenic region surrounding the eyes and contains all major lobes of the adult structure in miniature form[25,26] (see Fig. 1a–c). Upon hatching, the paralarval brain consists only of an estimated 200,000 cells[27], which makes it an attractive structure to build a cell-type atlas.

In this study, we report on cell-type diversity in the *O. vulgaris* brain at hatching. Comparing and combining single-cell and single-nuclei datasets, we systematically characterize 42 cell types within the brain and describe their transcriptomes. We spatially map several of these cell types with in situ hybridization and use cross-species comparisons to predict conserved cell types and compare gene expression signatures. We identify key transcription factors specifying cell types within the brain and provide evidence that several cell types display unique combinations of PCDH, ZnF, or GPCR, suggesting that octopus-specific gene expansions contributed to increased cell-type diversity. While we estimate the diversity of octopus brain cell types to be larger than our current view, our results are a valuable resource for future studies and offer insights into the molecular profile of octopus brain cells and the evolution of cell types.

## Results and discussion

### Generation of a single-cell and single-nucleus transcriptome atlas

We performed 10x Genomics single-cell RNA sequencing using both nuclei and cells from dissected brains of one-day-old *O. vulgaris* paralarvae (Fig. 1a–d, see also Methods). The genome annotation of the draft genome of *O. vulgaris* is very poor because of its fragmentary assembly[28]. Therefore, we mapped the reads to the chromosomal scale genome assembly of *Octopus sinensis*[29], a very closely related species to *O. vulgaris*[30]. We were able to map 80 and 88.4% of all reads to this genome, for the nuclei and cells, respectively. We improved the genome annotation of this assembly to further optimize the accuracy of gene expression counts. Since this single-cell RNA-seq method is biased towards 3′ ends of messenger RNAs, we improved the 3′ UTR annotation by integrating FLAM-seq[31] and Iso-Seq (PacBio) full-length mRNA sequencing data of embryonic, paralarval, and adult octopus tissue[25,32] (Supplementary Fig. 1 and Data 1). With this new annotation, the percentage of reads that mapped confidently to the transcriptome increased significantly (from 32.5 to 45.6% for the nuclei and from 49.4 to 58.8% for the cells; Supplementary Data 2). We obtained 8517 nuclei and 8564 cells that passed QC thresholds (on gene counts and mitochondrial reads, see Methods). The median number of genes detected was 1351 and 1506 for nuclei and cells, respectively. After batch effect correction, we combined these cells and nuclei into a single dataset containing 17,081 high-quality transcriptomes. This evidence-guided approach to annotate 3′UTRs and novel genes based on full-length RNA sequencing led to more reliable results and a higher number of estimated cells. Even in established model organisms such as zebrafish,

a similar approach led to the identification of additional cell types[33]. This method and resource (Supplementary Data 1) might aid other researchers in mapping bulk and scRNA-seq datasets.

We used a dual approach in sequencing cells and nuclei. By combining both datasets, we aimed to (1) increase the power of the analysis by analyzing more cells and therefore being able to identify more cell types and (2) identify and avoid technical artefacts introduced by a specific method. Almost all the clusters contained data points from both cells and nuclei (Supplementary Fig. 2c), which is a strong argument that we identified clusters that represent real cell types. Some cell types were less abundant in the nuclei (ACH1, ACH3, GLIA3, and Pep-burs), while others were underrepresented in the cells (FBL and GLIA2) (Supplementary Fig. 3). scRNA-seq captures more information (lowly expressed genes and cytosolic RNAs) but also introduces dissociation artefacts. We did find that heat shock proteins and immediate early genes were highly expressed in the cells, versus lncRNAs in the nuclei (Supplementary Fig. 3d). Immediate early gene *egr1* (early growth response 1) was highly expressed in the nuclei and might accurately indicate neuronal activation within the snRNA-seq data (Supplementary Fig. 3e)[34]. Varying clustering parameters, such as the number of principal components used, k-nearest neighbor, and cluster resolution resulted in different numbers of clusters and cluster sizes. Since previous knowledge of the expected number of cell types and their molecular markers was scarce, we assigned a stability value to each cluster in order to detect meaningful cell types (Supplementary Fig. 2b)[35]. By subsampling and reclustering the dataset, we identified the optimal clustering parameters that resulted in the highest number of stable clusters. Out of the 87 predicted clusters, 42 could be considered stable clusters (Fig. 1e). We found that this data-driven method[35] yielded biologically relevant clusters (see below) and was a reliable approach to discover new cell types.

We were able to sequence around 17,000 single-cell expression profiles, which is roughly 9% of the total number of cells in the paralarval brain[27]. The 42 stable clusters found are likely an underestimation of the total number of cell types present. To allow further exploration of this atlas by the community, we made it available as a portal in SCope (https://scope.aertslab.org/#/Octopus_Brain/).

### Cluster annotation based on neurotransmitter and peptide expression

The majority of cells present in the octopus paralarval brain were neurons (89% *elav*+, 83% *onecut*+, Supplementary Fig. 4a). Several neuronal types strongly exhibited a particular neurotransmitter or a peptidergic phenotype and were annotated accordingly, making use of gene homologs of fly and/or mouse (Fig. 2a). We could identify known cell types in the dataset such as dopaminergic (DOP), GABAergic (GABA), serotonergic (SERT) and peptidergic (PEP-*fmrfa*+) neuronal subtypes. A large body of work has been done on the localization of FMRFamide (*fmrfa*) synthesizing neurons in different cephalopods[17,36–39]. In this study, we identified three different *fmrfa* precursor genes which are differentially expressed between cell types (Fig. 2a). In addition, we identified a range of additional peptides, such as Crustacean cardioactive peptide *ccap*[40,41], which have not yet been described during cephalopod development. The majority of neurons expressed one or more neuropeptides, in addition to a neurotransmitter, e.g., DOP3; tyrosine hydroxylase (*th*) and *prqfva1*. In contrast, some clusters did not have a clear neurotransmitter phenotype but did express a prominent neuropeptide, for instance, *fmrfa3* (PEP-Fmrfa3) or *ccap* (CCAP) (Fig. 2a).

Furthermore, we show the presence of glutamatergic (vesicular glutamate transporter, *vglut*), cholinergic (vesicular acetylcholine transporter, *vacht*), and putative octopaminergic (tyramine beta-hydroxylase, *tbh*) neurons already at hatching, which were known to occur in adult cephalopod nervous systems (reviewed by ref. 42). The paralarval brain was mostly glutamatergic (64% *vglut*+) and cholinergic

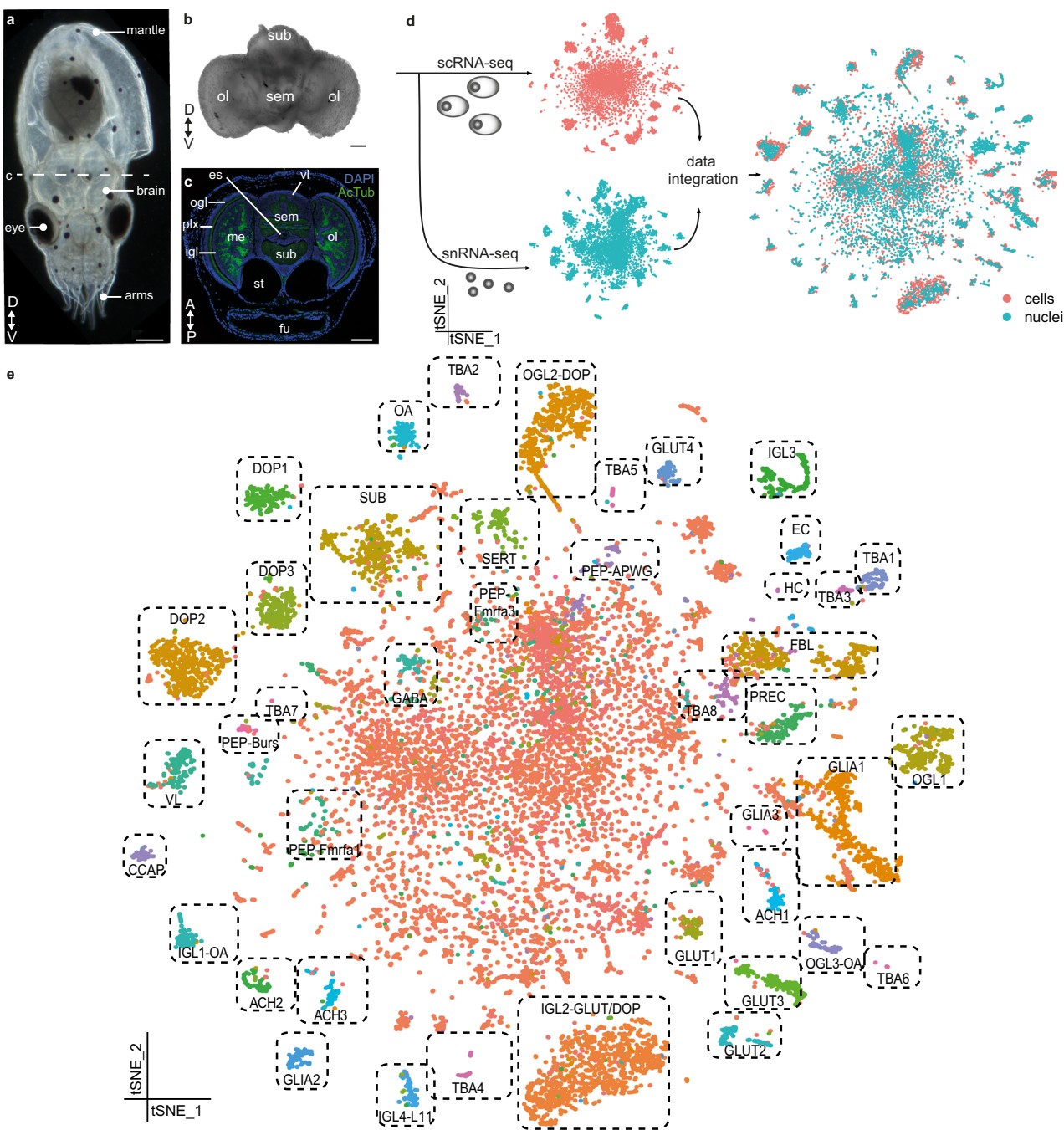

**Fig. 1 | Cellular diversity in the developing octopus brain. a** One-day-old *Octopus vulgaris* paralarva. The dashed line indicates the sectioning plane in **c**. **b** Anterior view of a dissected brain. **c** Representative transversal section of a larva with annotated anatomical structures. **d** Experimental design of this study. Single cell and nuclei RNA sequencing was performed with 10x Genomics and data integration resulted in a filtered dataset of 17,081 high-quality cells. **e** t-SNE representation of the integrated sc and snRNA-seq data. Annotated cell types are labeled. All scale bars represent 100 µm. A anterior, ACH cholinergic neurons, AcTub acetylated tubulin, CCAP cardioactive peptide cells, DOP dopaminergic neurons, D dorsal, EC endothelial cells, es esophagus, FBL fibroblasts, fu funnel, GABA GABAergic neurons, GLUT glutamatergic neurons, HC hemocytes, igl inner granular layer, IGL inner granular layer cells, me medulla, OA octopaminergic neurons, ogl outer granular layer, OGL outer granular layer cells, ol optic lobe, P posterior, PEP peptidergic neurons, PREC precursor cells, plx plexiform layer, sem supraesophageal mass, SERT serotonergic neurons, sub subesophageal mass, SUB subesophageal neurons, st statocysts, TBA to be annotated, V ventral, vl vertical lobe, VL vertical lobe cells.

(29% *vacht*+), yet we also found four prominent dopaminergic clusters (27% of all cells are *th*+) (Fig. 2a). Other molecules involved in neuromodulation in the adult octopus nervous system, such as noradrenaline, tyramine, histamine, substance P, somatostatin and VIP, were not identified[42,43].

On the t-SNE plot, a large central constellation of neurons was visible that we could not assign to a stable cluster. These cells could be divided into a cholinergic and glutamatergic population (Fig. 2b) and quality control metrics were similar for cells in the central constellation and cells in the periphery (Supplementary Fig. 5b). Based on immediate early gene expression in the nuclei (early growth response protein1, *egr1*), we identified a difference in cell states between the periphery (less active) versus the central constellation (more active). We found that transcription factors like dimmed (*dimm*) and rotund/squeeze

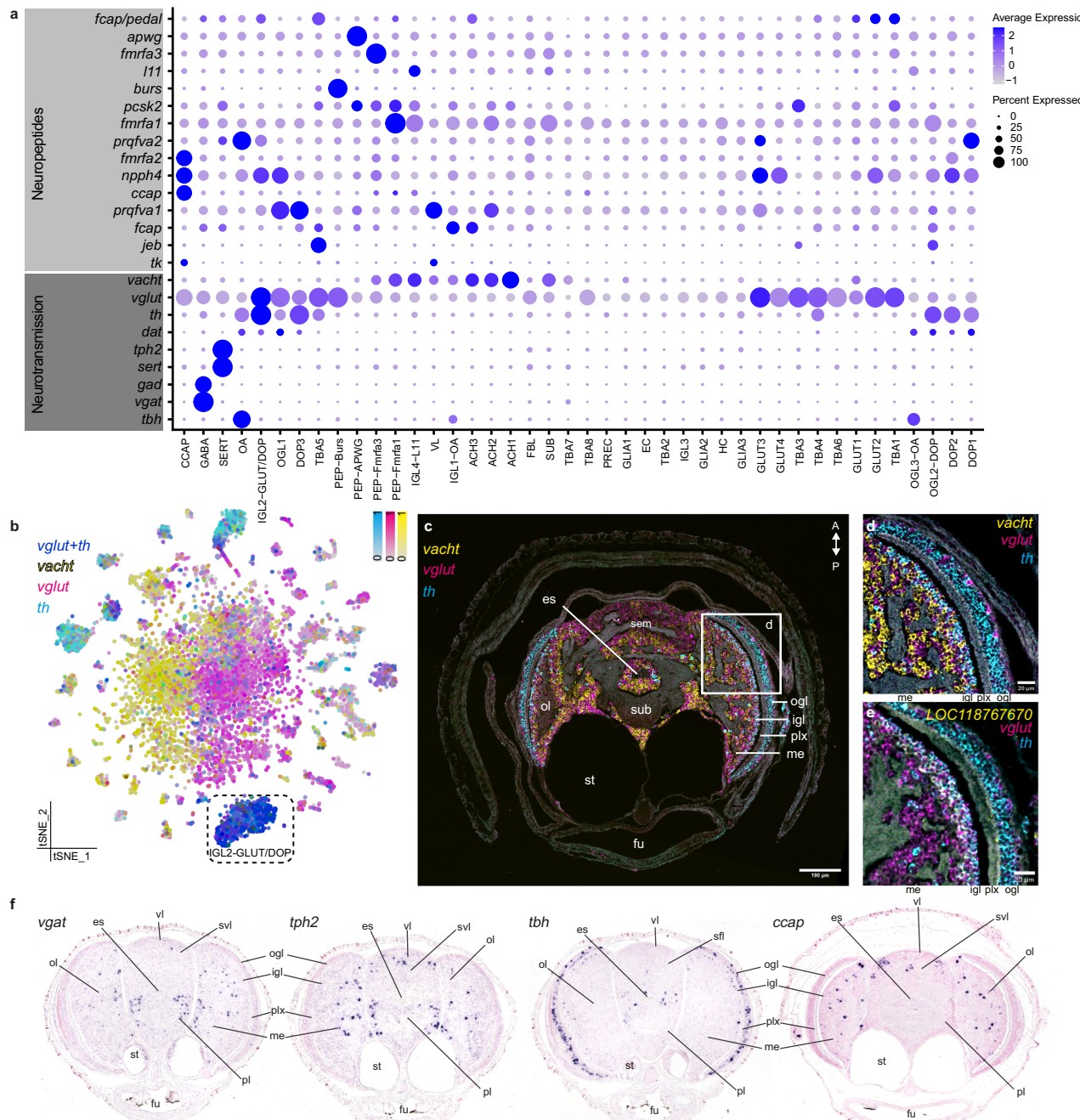

**Fig. 2 | Neurotransmitters and peptides. a** Dot plot for main neuropeptides and genes involved in the synthesis and transport of neurotransmitters. **b** Expression of *th*, *vglut*, and *vacht* is visualized on a t-SNE plot. *Th* (tyrosine hydroxylase) is shown in cyan, *vglut* (vesicular glutamate transporter) in magenta, and *vacht* (vesicular acetylcholine transporter) in yellow. **c** Multiplexed in situ hybridization chain reaction (HCR) for *th*, *vglut*, and *vacht*. The white square is shown in **d** at a higher magnification. **d** Multiplexed in situ hybridization chain reaction (HCR) in the optic lobe for *th*, *vglut*, and *vacht*. **e** Co-expression of *th* and *vglut* in the inner granular layer of the optic lobe, together with the cluster-specific marker for IGL2-GLUT/

DOP; *LOC118767670*. **f** In situ hybridization for *vgat* (vesicular GABA transporter, GABAergic neurons), *tph2* (tryptophan hydroxylase 2, serotonergic neurons), *tbh* (Tyramine β-hydroxylase, octopaminergic neurons), and the neuropeptide *ccap* (Crustacean cardioactive peptide). Scale bars represent 100 μm for the overview images. A anterior, es esophagus, fu funnel, igl inner granular layer, ifl inferior frontal lobe, me medulla, ogl outer granular layer, ol optic lobe, pl pedal lobe, plx plexiform layer, P posterior, sfl superior frontal lobe, st statocysts, svl subvertical lobe, vl vertical lobe.

(*rn/sqz*) were highly expressed within the central constellation, while transcription factor A, mitochondrial (*tfam*) and B-cell lymphoma/leukemia 11a/b (*bcl11a/b*) were enriched in the surrounding clusters (Supplementary Fig. 5c, d). Both *dimm* and *sqz* are important for the development of peptidergic cell types in *Drosophila melanogaster*[44,45]. To investigate whether cells in the central constellation were perhaps immature precursors, we characterized neuronal differentiation states based on their transcriptional diversity[46] (Supplementary Fig. 6).

Intriguingly, we found that dopaminergic and optic lobe cell types (DOP, IGL, and OGL) were predicted to be the most differentiated. While most cells in the central constellation show high transcriptional diversity and are thus predicted to be more immature, some cell types in the periphery (SUB, SERT, and IGL3) showed a similar profile.

The observation of a central constellation in the t-SNE was similar to what was seen in the *Drosophila* brain atlas[47]. A large set of neurons could not be clustered into distinct cell types even after sequencing

more than 60% of the total number of cells. These cells were spatially mapped to the central brain, which contains a large number of neuronal subtypes, each with a small number of cells[48]. To investigate whether this was also the case in our dataset, we spatially mapped central constellation cells using publicly available adult octopus bulk RNA-seq data of different brain areas[49]. We found that the central constellation is heterogeneous and contains cells that are transcriptionally similar to either the optic lobe or to the central brain (Supplementary Fig. 7a–c). Subclustering the central constellation revealed many smaller clusters surrounding a large central cluster (Supplementary Fig. 8a, b), supporting the hypothesis that it consists of a multitude of rare cell types. Intriguingly, we found that within the central constellation, neuropeptides are highly variable and are highly expressed in the smaller clusters (Supplementary Fig. 8c, d), in line with the higher *dimm* and *rn/sqz* expression. Taken together, our data suggested that the central constellation consists of rare cell types that are neuropeptidergic.

In situ hybridizations for highly expressed genes related to neurotransmitter synthesis or transport, and genes encoding peptides showed that glutamatergic and cholinergic cells were spread over the entire brain, while most dopaminergic cell types located in the optic lobe inner (igl) and outer granular layer (ogl) (Fig. 2c–f and Supplementary Figs. 9, 10). We identified a common dual-transmitter cell type (~5%), which is both dopaminergic and glutamatergic (Fig. 2b, e and Supplementary Fig. 10). In situ HCR in combination with the cluster-specific marker *LOC118767670* (uncharacterized membrane protein) showed that this cell type was prevalent in the igl of the optic lobe (IGL2-GLUT/DOP). Similarly, in the larval fly brain, 9% of neurons co-expressed markers for glutamatergic and aminergic neurons[50], while this cell type was less prevalent in the adult fly brain[51]. It remains to be determined whether these dual-transmitter neurons play a specific role during development.

GABAergic (glutamate decarboxylase, *gad*, and vesicular GABA transporter, *vgat*) and serotonergic (sodium-dependent serotonin transporter, *sert*, and tryptophan 5-hydroxylase 2, *tph2*) neurons comprised smaller populations (GABA and SERT) that were located throughout the medulla of the optic lobe and the central brain (Fig. 2f and Supplementary Fig. 4b). Octopaminergic cell types (OA) expressed the synthesizing enzyme *tbh* and were mainly located within the cortex of the optic lobe (Fig. 2f; OGL3-OA, IGL1-OA). *Ccap*+ cells were observed in the subvertical lobe and the optic lobe medulla (Fig. 2f). Although it was quite a surprise to find ecdysis-related neuropeptides, such as *ccap* and *bursicon*, in a lophotrochozoan brain, recent evidence from other species suggests that these might play a role in hatching processes[52]. We also identified a cholinergic cell type (SUB) corresponding to cells in the subesophageal mass (Supplementary Fig. 11). Based on their very specific location, we hypothesize that these might be the so-called pear-shaped "giant-cells" that are localized in the posterior lateral pedal lobe in octopods[6]. These neurons produced the neuropeptide elevenin (*l11*) and sonic hedgehog (*shh*) and were organized in groups of large cells within the sub. Furthermore, this cell type appeared intercalated with glutamatergic neurons (Fig. 2c and Supplementary Fig. 9). In *O. bimaculoides, shh* expression was also observed in the arms and in the pallioviseral cord (prospective sub)[13], where it might act as an important signal for neural patterning.

## Molecular lamination within the deep-retina

We further investigated whether neuronal subtypes were spatially confined or distributed, by mapping subtype-specific marker genes. In octopus, photoreceptor cells in the eye project directly to cells in the granular layers of the optic lobe, which is also called the 'deep-retina'[10,53]. We found three distinct cell types within the ogl of the optic lobe (Fig. 3a–d). Most cells in the ogl were small dopaminergic neurons (OGL2-DOP). Different cells in the OGL2-DOP cluster expressed neuropeptide jelly belly (*jeb*) and down-syndrome cell adhesion molecule

(*dscam*), suggesting molecular subtypes. *Dscam*+ cells were located more towards the interior of the layer, while *jeb*+ cells were positioned more externally (Supplementary Fig. 12). *Dscam*, required for the proper lamination of amacrine cells in the mouse retina[54], might have a similar role in the octopus optic lobe cortex. Cell bodies of a second cell type (OGL1) seemed slightly larger than the dopaminergic cells and did not have a prominent neurotransmitter/peptidergic phenotype. These cells specifically expressed Protein phosphatase 1 (*ppp1*). The largest cell bodies we identified were octopaminergic, expressed protocadherin O2 (OGL3-OA), and were a lot less prevalent than OGL1 and OGL2-DOP. OGL1-3 likely represent the differentially sized unipolar amacrine cells identified in the adult ogl by Young[10]. Furthermore, we observed multiple cell types within the igl (Fig. 3e–h). Large epidermal growth factor receptor-positive cells (*egfr*+, IGL1-OA) were located externally, next to the plexiform layer and are octopaminergic. Conversely, StAR Related Lipid Transfer Domain-containing 5 (*stard5*+) cells (IGL4-L11) and *calbindin* + cells (IGL3) were organized in layers more towards the medulla. In vertebrates, *calbindin* is expressed in cones, bipolar, and amacrine retinal cells[55]. Intriguingly, IGL3 cells did not synthesize any prominent neurotransmitters or neuropeptides. A fourth glutamatergic and dopaminergic igl population (IGL2-GLUT/DOP) has been discussed above (Fig. 2e and Supplementary Fig. 10). Taken together, the optic lobe seemed more differentiated compared to other brain regions (Supplementary Fig. 6). We found multiple cell types that further divide the ogl and igl into molecularly distinct sublayers, similar in composition to those identified in *O. bimaculoides*[56]. This laminated appearance of molecularly different cell types is reminiscent of the vertebrate retina, as well as the optic lobe medulla in the fly[57,58].

## Cross-species cell-type comparisons

In order to identify and annotate evolutionarily conserved cell types, we performed comparisons between octopus, fly[47], and mouse[59] brain single-cell datasets using the SAMap algorithm[60] (Fig. 4a). The majority of octopus cell types did not have a predicted homologous cell type in fly or mouse. Conversely, we found that the octopus GLIA1 subtype is molecularly similar to fly ensheathing glia and mouse astrocytes, and GLIA3 to mouse telencephalic astrocytes (Fig. 4a). We could also deduce a conserved octopus-fly-mouse glial gene expression signature containing one-to-one orthologs of common glial genes (Fig. 4b and Supplementary Fig. 13). The existence of a common glial gene set between members of Lophotrochozoa, Ecdysozoa, and chordates, suggests that those genes reflect an ancestral bilaterian expression signature. Although it was suggested that glial cells likely evolved multiple times during evolution[61], our results might support the existence of a urbilaterian glial cell type.

While there are generally more glial cells in the mammalian brain than neurons, the opposite is true for most invertebrate species. Only around 10% of all cells in the octopus paralarval brain were identified as glia (glutamine synthetase 2, *gs2*+), see Fig. 4c. Both *gs2* and apolipoprotein (*apolpp*) were highly expressed in all glial populations. We found that many glial cells, including cells with multiple processes, were located in the neuropil near the axons of the cells from the perikaryal layer (Fig. 4d–g). Some glial cells were located between the neuronal cell bodies (Fig. 4f). Considering their unmyelinated and large central nervous system, cephalopods needed to develop alternative strategies to ensure conduction speed, e.g. the famous giant axon in squids[62]. Aside from myelin-producing glia in vertebrates, wrapping glia in *Drosophila* insulate axons and contribute to increased signaling speed[63].

We observed high expression of several invertebrate glial markers such as *CG6216* and excitatory amino acid transporter 1 (*eaat1*), but no orthologues could be identified for genes used to discriminate between glial subtypes in flies, i.e., I'm not dead yet (*indy*), astrocytic leucine-rich repeat molecule (*alrm*) and *wrapper*[47,50]. At least three

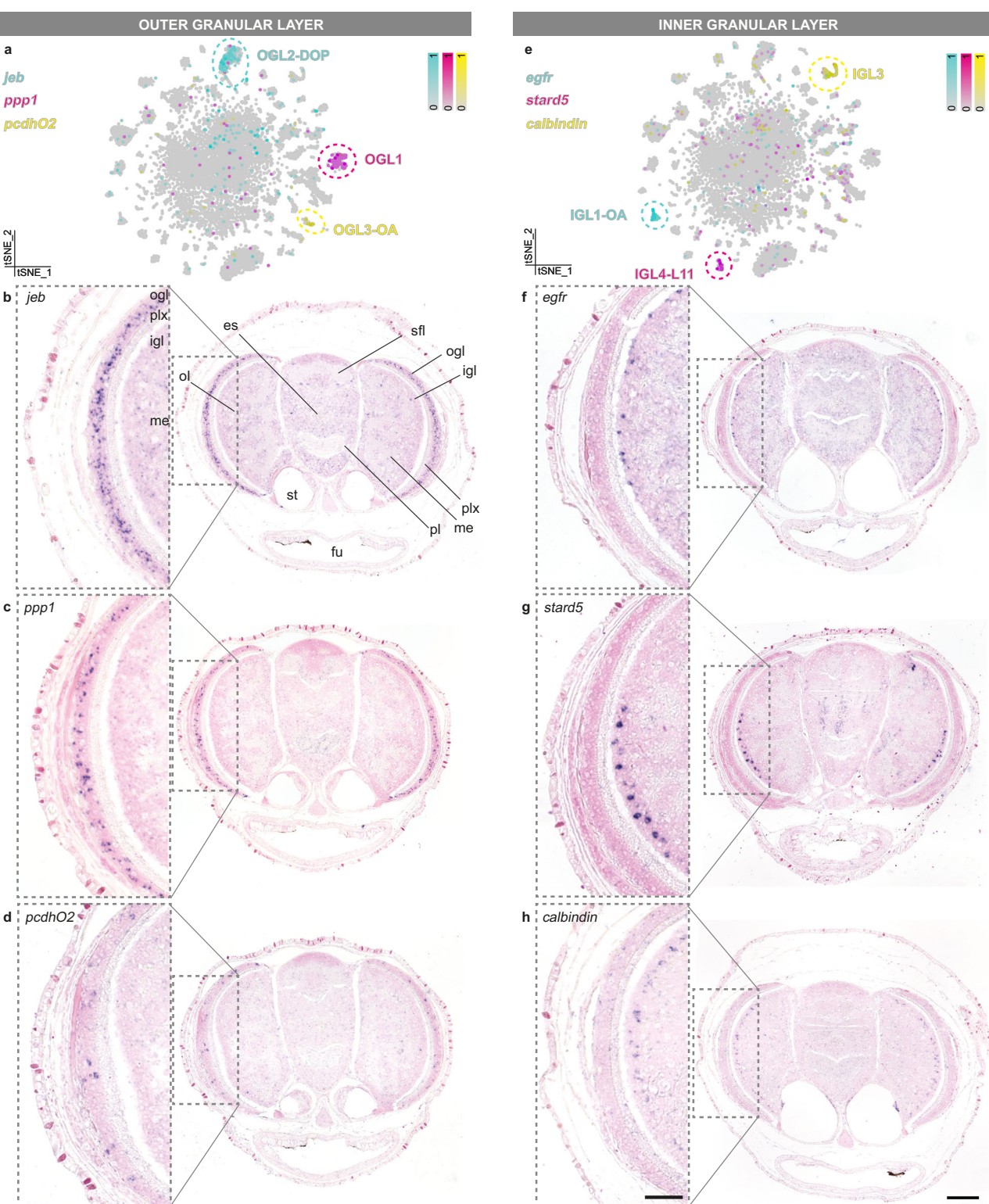

**Fig. 3 | Optic lobe cell-type diversity. a** Marker expression of outer granular layer cell types. **b** *jeb* is expressed throughout dopaminergic neurons in the ogl. **c** *ppp1* is expressed within the ogl, in fewer cells than *jeb*. **d** *pcdhO2* is expressed in ogl octopaminergic neurons. **e** Marker expression of inner granular layer cell types. **f** *egfr* is expressed in the outer region of the igl. **g** *stard5* is expressed more interiorly within the igl than the *egfr* + cells. **h** *calbindin* is expressed in the most interior side of the igl. In situ hybridizations are shown in **b**–**d**, **f**–**h**. Gray boxes indicate magnified regions. Scale bars are 100 μm for overviews and 50 μm for magnifications. es esophagus, fu funnel, igl inner granular layer, me medulla, ogl outer granular layer, ol optic lobe, pl pedal lobe, plx plexiform layer, sfl superior frontal lobe, st statocysts.

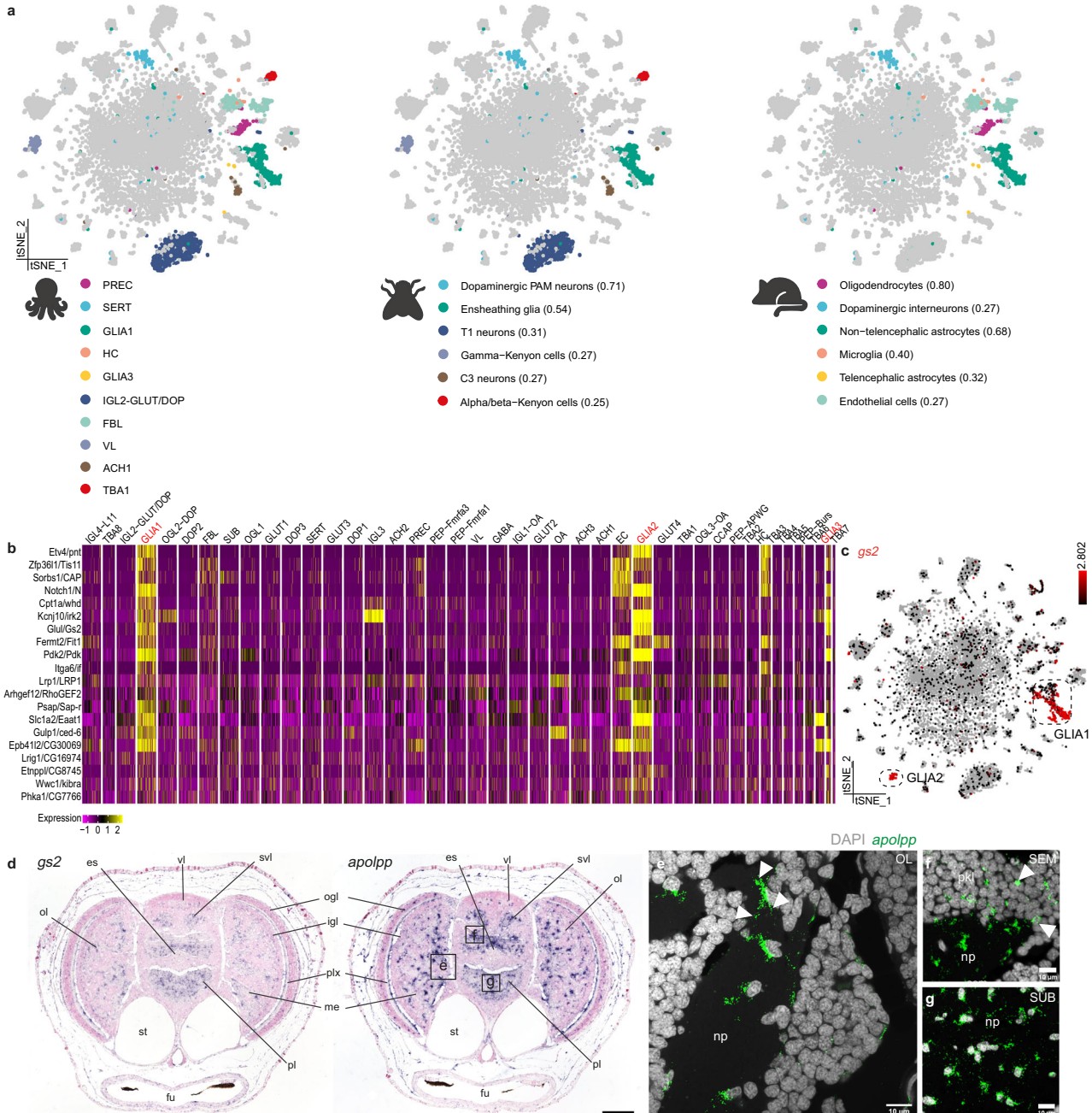

**Fig. 4 | Cross-species cell-type comparisons identify a glial gene expression signature. a** Cell-type mappings between octopus, fly, and mouse are represented on the t-SNE plot. Mappings are color-coded and alignment scores are shown between brackets. **b** Best reciprocal blast hits from the mapping between octopus GLIA1, fly ensheathing glia, and non-telencephalic astrocytes in the mouse brain. Glial populations are highlighted in red. **c** t-SNE representation of two main glial populations in octopus based on the expression of *gs2* (Glutamine synthetase 2). **d** In situ hybridization of *gs2* and *apolpp* (Apolipoprotein). Scale bar represents 100 μm. Representative magnifications of different brain areas are annotated with black boxes. Fluorescent in situ hybridization for *apolpp* are shown in **e**–**g**. **e** Glial cells in the optic lobe. White arrows indicate multiple processes. **f** Glial cells in the supraesophageal mass. White arrows indicate infiltrating glia. **g** Glial cells in the subesophageal mass. es esophagus, fu funnel, igl inner granular layer, me medulla, np neuropil, ogl outer granular layer, pkl perikaryal layer, ol optic lobe, pl pedal lobe, plx plexiform layer, SEM supraesophageal mass, st statocysts, SUB subesophageal mass, svl subvertical lobe, vl vertical lobe.

distinct glial subtypes were identified within this dataset (GLIA1,2,3), suggesting there is also functional diversification within octopus glial cells (Supplementary Fig. 14). The largest glial cluster identified (GLIA1) mainly expressed GABA transporter 1 (*gat1*), which was also found in neuropil glia or astrocytes in the fly brain[50]. To distinguish GLIA1 from GABAergic cells and the other GLIA types, we used a combination of *gat1 and apolpp* HCR. GLIA1 (*gat1+/apolpp+*) resemble astrocytes (multipolar) and were localized within the neuropil (Supplementary Fig. 15d–f). Based on the location of GLIA1 within the octopus brain, we

can differentiate between neuropil glia which have many processes, presumably involved in axon wrapping, and a small subset of infiltrating glia (Fig. 4f). The infiltrating glia likely provide support (both structural and metabolic) and might be involved in neuronal modulation as has been described for vertebrate astrocytes[64], since they are in close proximity to the neuronal cell bodies. *Gat1+/apolpp−* marked GABAergic neurons (Supplementary Fig. 15g–i). *Gat1−/apolpp+* cells (GLIA2/3) appeared as round-shaped nuclei in the plexiform layer of the optic lobe and elongated nuclei in a membrane-like layer that

surrounds the brain (Supplementary Fig. 15j–l). The plexiform layer that is located between the ogl and igl of the optic lobe contains mainly neurites but also glial cells with radial and tangential processes have been described[10]. In addition, the membrane-like layer that surrounds the brain lobes is reminiscent of the neuroglial folds that resemble the subpial astrocytic layer in vertebrates[65]. GLIA2 differentially expressed adhesion molecules such as *dachsous* and *cadherin23* and neuro-transmitter receptors such as metabotropic glutamate receptor 3 (*grm3*) and a *slc6a2* transporter (Supplementary Fig. 14). Conversely, GLIA3 expressed vascular endothelial growth factor receptor (*vegfr*), and very few genes that were involved in neuromodulation. We, therefore, hypothesize that GLIA2 are the glia in the plexiform layer and that GLIA3 form a membrane surrounding the brain (not directly in contact with the neurons), and might play a role in the hemolymph-brain-barrier, similar to astrocytes. Both GLIA1 and GLIA3 mapped to mouse astrocytes, while GLIA2 was not identified in the cross-species mappings (Fig. 4a).

SAMap also found similarities between octopus serotonergic neurons, fly dopaminergic PAM neurons, and dopaminergic inter-neurons in mice (Fig.4a). In addition, fly lamina feedback C3 neurons map to the octopus cell-type ACH1. IGL2-GLUT/DOP was a prominent cell type within the octopus visual system and had a similar molecular profile to fly T1 neurons (e.g., *eaat1* and gamma-interferon-inducible lysosomal thiol reductase 1, *gilt1*) (Fig.4a). It remains to be investigated whether IGL2-GLUT/DOP neurons are the amacrine cells that provide feedback from the igl to the plexiform layer, similar to the fly T1 neurons from the medulla to the lamina. To identify on which genes the homology hits were based, gene lists were sorted based on high expression in octopus and the top five reciprocal blast hits together with transcription factors with a good reference orthologue were plotted for each cell type for fly and mouse, respectively (Supplementary Figs. 16, 17). We find that each putative homology (except for TBA1/ABKC) relied on several transcription factors which have good reference orthologs in mouse/fly.

Another interesting observation from the SAMap comparison was the similarity between octopus vertical lobe cells (VL) and fly gamma Kenyon cells (γ-KC) (Fig. 4a). The vertical lobe (vl) is considered to contribute to learning and memory in octopus[7,66–68]. After ablation experiments of the vl, memory formation was found to be impaired[67]. Based on its 'fan-out fan-in' matrix-like synaptic organization, its folded anatomy, small interneurons and the existence of LTP[69], this structure is suggested to be functionally analogous to the insect mushroom body[5,11]. Mushroom body-like structures have been identified in other lophotrochozoans such as *Platynereis* and a common origin has been suggested[70,71]. VL marker genes *aristaless* (*arx*), cAMP-dependent protein kinase regulatory subunit type II (*pka-R2*), and transmem-brane *O*-mannosyltransferase targeting cadherins 4 (*tmtc4*) were widely expressed in the vl (Fig. 5a–c), and colocalized with *vacht* expression, suggesting that these cells are the cholinergic amacrine cells described in the adult vl[66]. Gene ontology enrichment analysis showed that cognition, learning, and learning or memory are the top three most enriched biological processes. This further supports the function of the vl as the structure contributing to associative proces-sing and learning and memory in the octopus brain[67]. Regarding the molecular profile of these cells (Fig. 5d), genes involved in long-term potentiation and memory formation, such as components of the cAMP/PKA pathway[72] (e.g., *pka-R2*, *rutabaga*), were highly expressed. Common marker genes identifying the mushroom body in the fly, like Dunce (*dnc*) and Leo (*pka-c*)[73], were also enriched within VL. Con-servation of gene expression profiles might point towards a common origin (out of an ancestral cell type) or a common function (by means of convergent evolution). Certain transcription factors, i.e. myocyte enhancer factor 2 (*mef2*), mushroom body specific/ecdysone-induced protein 93 F (*mblk/eip93f*), dorsal switch protein1 (*dsp1*) and zn finger homeodomain 2 (*zfh2*) were present in both VL and Kenyon cells

(logfc.threshold >0.25). However, there was no significant enrichment for typical Kenyon cell transcription factors such as *eyeless, fruitless* and *datilografo*[47,74]. Moreover, other transcription factors that have been commonly found in anterior brain structures in *Platynereis* and vertebrates, such as paired-box protein (*pax6*), Homeobox protein *emx2*, LIM/Homeobox protein *lhx6* or Homeobox protein *nkx2-1*[75], were not found to be expressed in the VL. On the other hand, the VL highly expressed *arx*, which is a central transcription factor demar-cating the early mushroom body in the annelid *Platynereis*[70]. In addi-tion, the VL-γKC mapping was dependent on *mef2*, which is important for mushroom body development[76], and *zfh2*, which is expressed in a Kenyon cell subtype in *Drosophila*[77] (Supplementary Fig. 17). These findings suggest that some cell types might deploy deeply conserved transcriptional programs across bilaterian evolution.

## Non-neuronal cell types

Considering that this is a paralarval brain, of which the number of cells still needs to multiply a thousand-fold to reach adulthood[9,27], the diversity of mature neuron types is impressive. Aside from a relatively small precursor population (~1%), most cells are post-mitotic. In a previous study, we identified the lateral lips as the neurogenic niche outside of the developing octopus brain[25]. The lateral lips are anatomically very closely connected with the central brain through the anterior and posterior transition zones. We could retrieve limited expression of previously identified transcription factors achaete-scute homolog 1 (*ascl1*) and neurogenic differ-entiation factor 1 (*neurod*), which we assumed were lateral lip/tran-sition zone cells (Supplementary Fig. 18). These precursors (PREC) highly expressed markers related to pluripotency, embryonic stem cells, and the npBAF complex. Genes such as insulinoma-associated protein 2 (*insm2*), *rootletin* and proliferation marker protein *ki67* were highly expressed within the precursors. The majority of pre-cursor cells were post-mitotic (*neurod*+), but a smaller population was still progenitor-like (*ascl1+*) (Supplementary Fig. 18b). Common markers for S and G2/M phase were highly expressed in this cluster (Supplementary Fig. 18c). At this stage, we could only find a minor population of proliferating cells (phosphorylated histone h3, PHH3+), within the remnants of the lateral lips but not in the brain (Supplementary Fig. 18d). SAMap found these precursors to be related to mouse oligodendrocytes (Fig. 4a). The resemblance with mouse oligodendrocytes might point to a common ancestral glial cell that has neural progenitor features. Important to note is that the SAMap predictions are dependent on one-to-many orthologue annotations and on the cell types present in both datasets. The adult fly brain and adult mouse telencephalon datasets used for compar-ison here might not have contained many neural progenitors making the retrieval of related cell types difficult. Future comparisons with datasets of younger life stages or more closely related species might reveal more similarities. Also note that while this paralarval brain represents the end point of embryonic neurogenesis, a secondary phase of neurogenesis during a later stage is likely to occur.

Contrary to most invertebrates, the octopus has a closed circu-latory system and a hemolymph-brain barrier[78–80]. At this develop-mental stage, we expected a certain degree of cerebral vasculature[81]. We found octopus endothelial cells (EC) that highly expressed con-served markers, more specifically *vegfr*, *troponin T*, developing brain Homeobox/H2.0-like Homeobox ortholog (*dbx/hlx-like*) and *meox2*, Homeobox protein mox-2 (see below). Furthermore, we identified a small population of hemocytes (HC) within the dataset that expressed vascular endothelial growth factor (*vegf*) and sushi, von Willebrand factor type A, EGF, and pentraxin domain-containing protein (*svep1*). We also observed high *vegf* expression underneath the epidermis in a punctuate pattern (Supplementary Fig. 19). The resemblance of the octopus hemocytes with mouse microglia, which are derived from the blood lineage, was not unexpected (Fig. 4a).

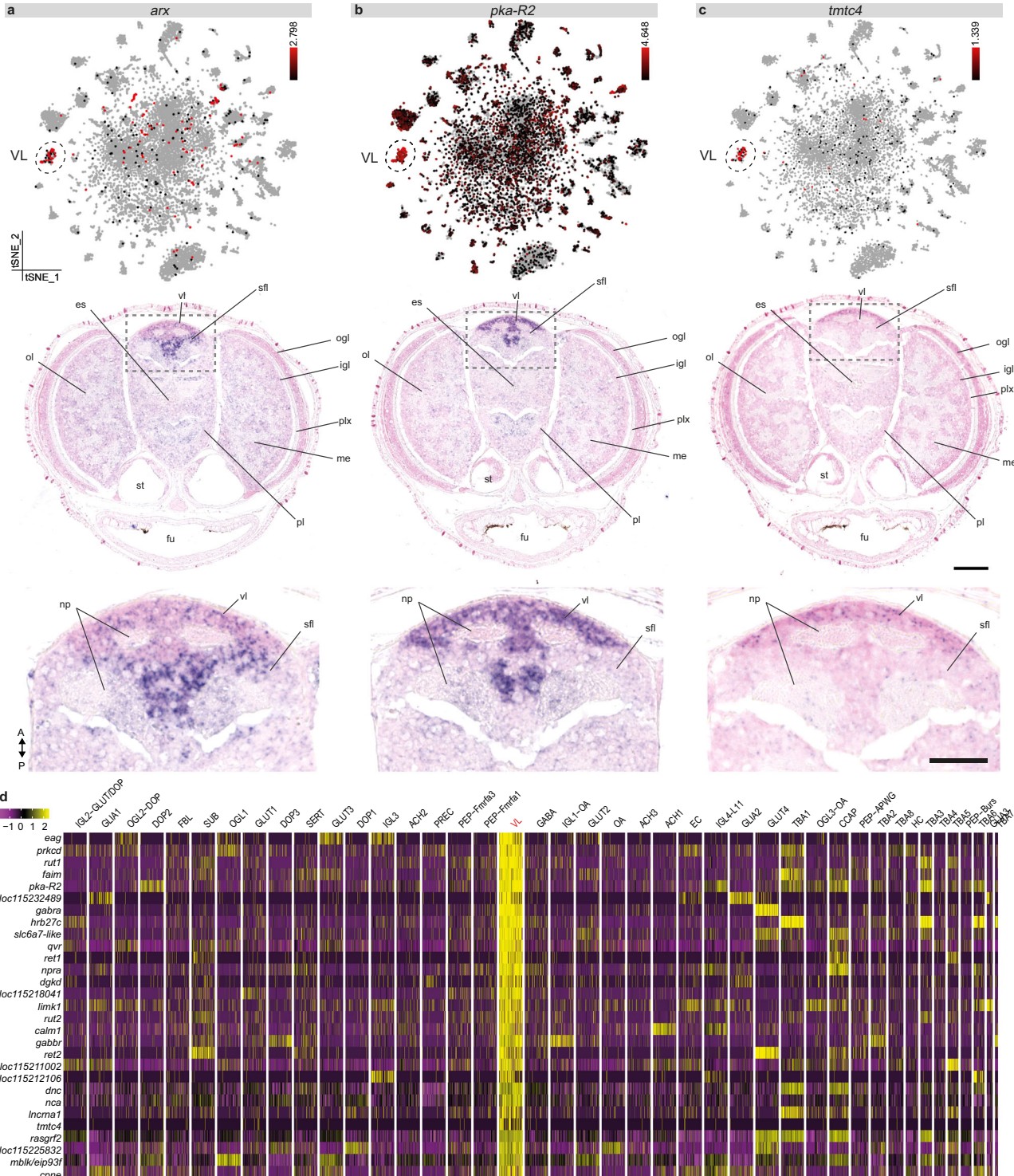

**Fig. 5 | The molecular profile of the vertical lobe cells. a** Aristaless (*arx*) expression is limited to the vertical (vl) and the superior frontal lobe (sfl). **b** cAMP-Dependent protein kinase regulatory subunit type II (*pka-R2*) expression can be observed mostly in the vl and less within the sfl. **c** Transmembrane O-Mannosyltransferase Targeting Cadherins 4 (*tmtc4*) is uniquely expressed within the most anterior part of the vl. t-SNE plots for *arx*, *pka-R2*, and *tmtc4* are shown in **a**–**c** together with their respective in situ hybridization. **d** Top 30 genes with the highest fold change for the vertical lobe cells (VL). Scale bars are 100 μm for the overview images and 50 μm for the magnifications. es esophagus, fu funnel, igl inner granular layer, np neuropil, me medulla, ogl outer granular layer, ol optic lobe, pl pedal lobe, plx plexiform layer, sfl superior frontal lobe, st statocysts, vl vertical lobe.

Fibroblast-like cells (FBL) were annotated based on their expression of collagens, troponin, tropomyosins, and ribosomal genes. Octopus fibroblasts were organized in a layer that surrounds the brain (Supplementary Fig. 19). As this cell type produced an extracellular matrix, it might contribute to forming the protective structure surrounding the central brain. Only half of the FBL expressed *troponin T* marking fully differentiated cells. Octopus FBL mapped to mouse endothelial cells, reflecting their common mesodermal origin (Fig. 4a).

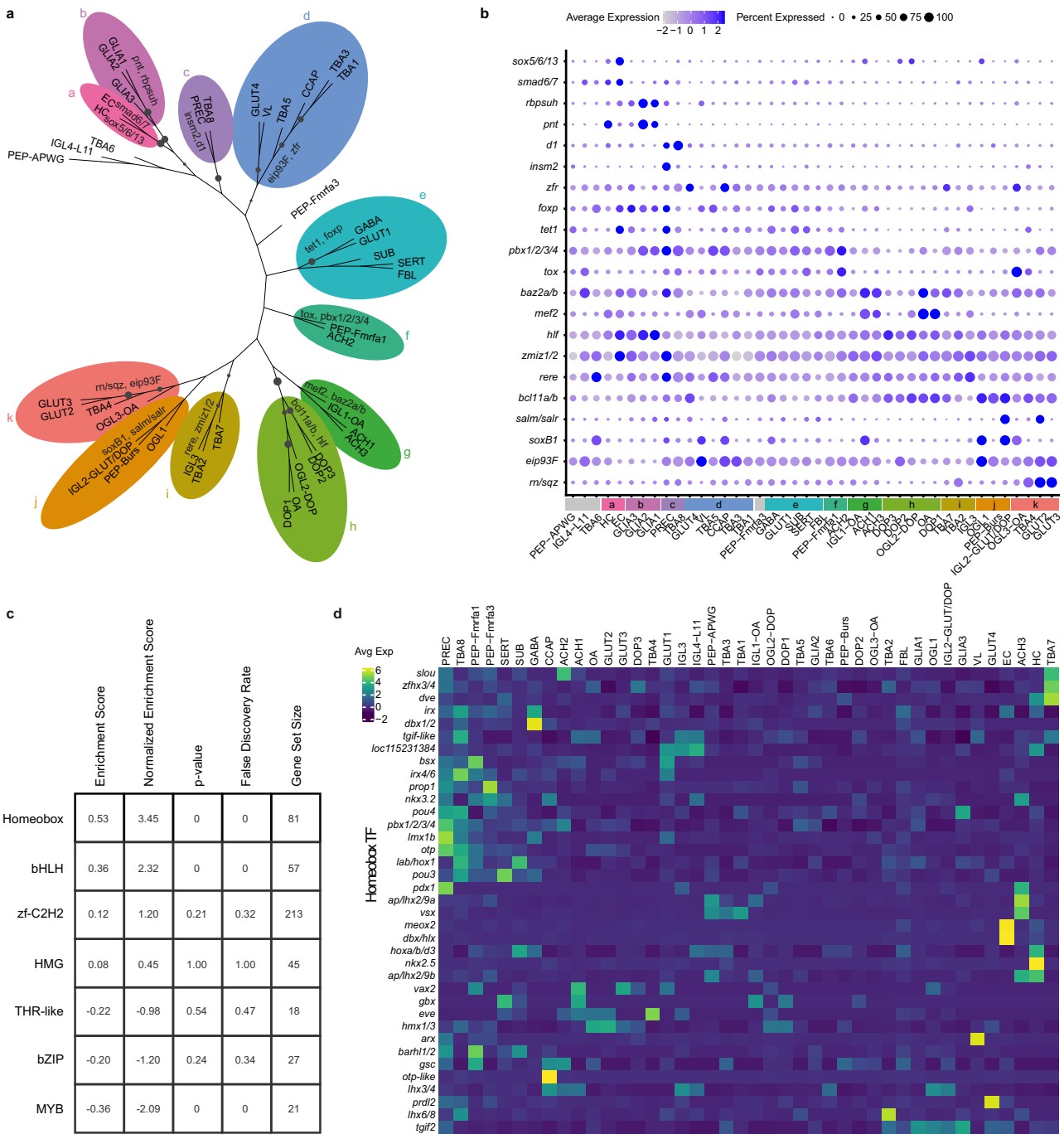

**Fig. 6 | Cell-type specificity and transcription factors. a** Neighbor-joining tree of octopus cell types. Branches with bootstrap support higher than 0.5 are visualized by increasing dot size. Differentially expressed transcription factors (TFs) for each group are shown. **b** Differentially expressed TFs from **a** are shown in a dot plot.

**c** Enrichment scores for the TF classes within the ranked list based on tau. **d** Heatmap of highly variable Homeobox transcription factors, averaged per cell type and scaled.

## Homeobox genes are defining transcription factors for cell-type identity

To further understand cell-type relationships, we built a cell-type taxonomy based on all genes (Fig. 6a). Most relationships were as expected (blood-glial-neuronal), except for FBL, which clustered with neuronal subtypes. Next, we sought to identify shared transcriptional programs by examining which transcription factors (TFs) were differentially expressed between the different branches of the tree. After subsetting for TFs, we identified the top differentially expressed TFs (with a reference orthologue in either mouse or fly) for each group against all others. Groups are color-coded on the tree and the

expression of the identified TFs are shown in a dot plot (Fig. 6a, b). We found that HC and EC were characterized by high expression of *sox5/6/13*, which confirms previous observations of *sox5/6/13* in the circulatory system of *Sepia officinalis*[82]. GLIA1,2 and 3 were specified by the ETS-TF pointed (*pnt*), similar to glial cells in *Drosophila*[83]. Regarding the specification of neuronal subtypes, we found that *soxB1* was differentially expressed in a subset of neurons (Fig. 6a). Previous work has shown that also in *O. vulgaris*, *soxB1* was expressed in post-mitotic neurons and is likely involved in neural differentiation[25]. Most dopaminergic cell types (DOP1, DOP2, DOP3, and OGL2-DOP) expressed *bcl11a/b*, a ZnF TF gene duplicated in mammals and key for the

development of a subset of mouse midbrain dopaminergic neurons (Bcl11a)[84]. Another branch grouped cell types such as GLUT4, VL, and CCAP (branch d), which are all localized in the (sub)vertical lobe (Figs. 2, 5 and Supplementary Fig. 20). These cell types highly expressed *eip93F* and might have a common origin. Although we did not spatially map all the cell types in the tree, there seemed to be a trend that optic lobe cell types (branch g-k) and central brain cell types (branch d-f) group together. Taken together, we identified the major TFs (in Fig. 6a, b) which are important for cell-type specification within the octopus brain. In our taxonomy, more supported branches seemed to contain cell types that were spatially close together in the brain, suggesting that in different regions, cell diversification might have happened from an ancestral cell type.

Recent studies in *C. elegans*[85] and in fly motoneurons[86] suggest that unique combinations of Homeobox TFs are responsible for maintaining cell-type identity. To investigate which TF families determine cell-type identity in octopus, we calculated the tissue specificity index (tau) for all TFs, which resulted in a ranked list. A gene set enrichment analysis for the different TF families within the ranked list identified the most cell type-specific one. We found that Homeobox TFs are the most linked with cell-type identity, followed by basic helix-loop-helix TFs and ZnF, which massively expanded in coleoid cephalopods (Fig. 6c). Combinations of Homeobox TFs do seem to be uniquely expressed in certain cell types (Fig. 6d), which suggests that the concept of a cell type determining Homeobox code is translatable to organisms with increased neuronal cell-type diversity. Besides *arx* (Fig. 5), highly expressed in the amacrine cells of the developing vertical lobe, we mapped the expression patterns of two other Homeobox TFs (*vsx* and *prdl2*) that very clearly delineated cell types. We found that the visual system Homeobox gene (*vsx*, Supplementary Fig. 20) is expressed mostly in the optic lobe medulla and in a few cells in the subesophageal mass, in line with what has been described in *S. officinalis*[87]. *Prdl2*, a paired-like Homeobox domain gene, clearly marked the GLUT4 population and was located in a very specific area in the subvertical lobe (Supplementary Fig. 20). This gene is the ortholog of *Doryteuthis pealeii prdl2*, which marked a distinct region of the cerebral cord (prospective subvertical lobe) in the developing squid embryo[88]. We found homology between the expression of the Hox gene labial (*lab*) in SUB (Fig. 6d) and expression in *Euprymna scolopes*[89] and *O. bimaculoides*[13] palliovisceral and pedal cords, embryonic structures that form the subesophageal mass. *Dbx1/2* was highly expressed in the GABAergic neurons, similar to *Dbx* expression in *Drosophila* which is restricted to GABAergic interneurons with short axons[90]. We can also observe specific expression *dbx/hlx* and *meox2* in EC, which are known regulators in human endothelial cells[91–93]. We find that hemocytes are characterized by a high expression of *nkx2.5*. This observation is consistent with previous research in squid[94] and cuttlefish[95], where high expression of *nkx2.5* in mesodermal structures such as the systemic heart was found. Taken together, Homeobox TFs expression was found to be conserved on numerous occasions, although we did not yet systematically map all clusters spatially.

### Genetic novelty drives cellular diversification

Our data showed a large diversity in brain cell types, which is expected in an animal with a rich cognitive behavioral pattern. Novel genes have been found enriched in species-specific cell types[96]. These genes can contribute novel features and lead to the evolution of unique cell types. Previous genomic studies indicated that coleoid cephalopods, including *O. bimaculoides* and *O. vulgaris*, specifically expanded certain gene families, leading to novel octopus genes[19,97]. We hypothesized that recently expanded gene families, such as PCDH, ZnF, and GPCR, might convey the potential to diversify and develop octopus-specific cell types. For this purpose, we investigated whether genes of these families are enriched in certain cell types, which could be considered as a metric for novelty (p-adj<0.05, based on Fisher's exact

tests, Bonferroni corrected; Supplementary Fig. 21). While distinct subsets of GPCR were highly expressed in specific neuronal cell types (GABA, SERT, and PEP-APWG), the ZnF were enriched in the precursor cells, pointing to a potential role in cell fate specification and differentiation. This corroborates the finding that ZnF genes are more highly expressed during embryogenesis in *O. bimaculoides*[19].

PCDH were often annotated as marker genes for specific cell types (logfc.threshold >0.25). We found that some PCDH were ubiquitously expressed, while others were enriched in specific cell types (Fig. 7a). *PcdhO1* (Fig. 7b) was highly expressed within serotonergic neurons (SERT), whereas *pcdhO2* (Fig. 7c) was enriched in a subset of octopaminergic neurons in the ogl (OGL3-OA). Important to note that these PCDH have evolved independently and hence are not orthologous to vertebrate PCDH. We grouped the PCDH based on their genomic location and identified 159 clustered and 13 non-clustered PCDH (located on different scaffolds). We could not identify any clear differences in expression based on their genomic location (Supplementary Fig. 22). Although neuronal and non-neuronal cell types express multiple PCDH genes (Supplementary Fig. 22a), the average expression of all PCDH is lower in non-neuronal cell types such as EC, GLIA, HC (Supplementary Fig. 22b). Based on raw counts of individual cells, we find that on average each cell expresses 19 different PCDH genes (Supplementary Fig. 22c). For neuronal cells, this number is higher (21 PCDH) than for non-neuronal cells (13 PCDH), suggesting that PCDH are more important in neurons, and might contribute to cellular diversification.

### Towards a single-cell view of the octopus brain

Here we provided an initial view of cell-type diversity of a highly complex invertebrate brain, which we have only begun to explore (Supplementary Figs. 23, 24). Based on the work by J.Z. Young, the number of real cell types likely ranges between 100–150 in the adult octopus nervous system[6,9]. Using the difference in nuclear size and anatomical location inferred from classical Golgi stainings as a metric, one can distinguish 116 different cellular phenotypes. In the hatchling brain, we were able to identify only 42 cell types out of the putative 116 present in the adult. Since this developing octopus brain still needs to grow, the number of cell types will also likely increase with age. Moreover, a large number of cells in the central constellation likely comprises many rare cell types. Increasing the number of sequenced cells might reveal additional heterogeneity of the central constellation and will likely resolve more cell types in future studies. Nevertheless, the dataset presented here already provides a starting point for comparative studies with other cephalopod species and/or with the adult octopus brain, which might yield informative answers linking brain complexity and cell-type diversity. Comparative studies that incorporate cell-type atlases of a more diverse range of invertebrate organisms can further elucidate cell-type homologies by examining transcription factor conservation. Multimodal analyses which enable evolutionary comparisons between gene regulatory networks might provide additional support and shed light on cell-type evolution. It remains an open question whether larger nervous systems also have more cell types or whether they have an increased cell number per cell type. Larger cell numbers of certain cell types might increase the computational power of the brain, which could explain the higher cognitive function of the octopus brain.

## Methods
### Genome annotation
The chromosomal scale genome assembly for *Octopus sinensis* was used (ASM634580v1, https://www.ncbi.nlm.nih.gov/data-hub/genome/GCF_006345805.1/)[29]. We extended the 3′-ends of the genes using an evidence-guided approach (https://github.com/rajewsky-lab/octopus_microRNAs/tree/main/gene_extension). First, full isoform-sequencing data (Iso-Seq, PacBio Sequel) was used to reconstruct mRNA isoforms

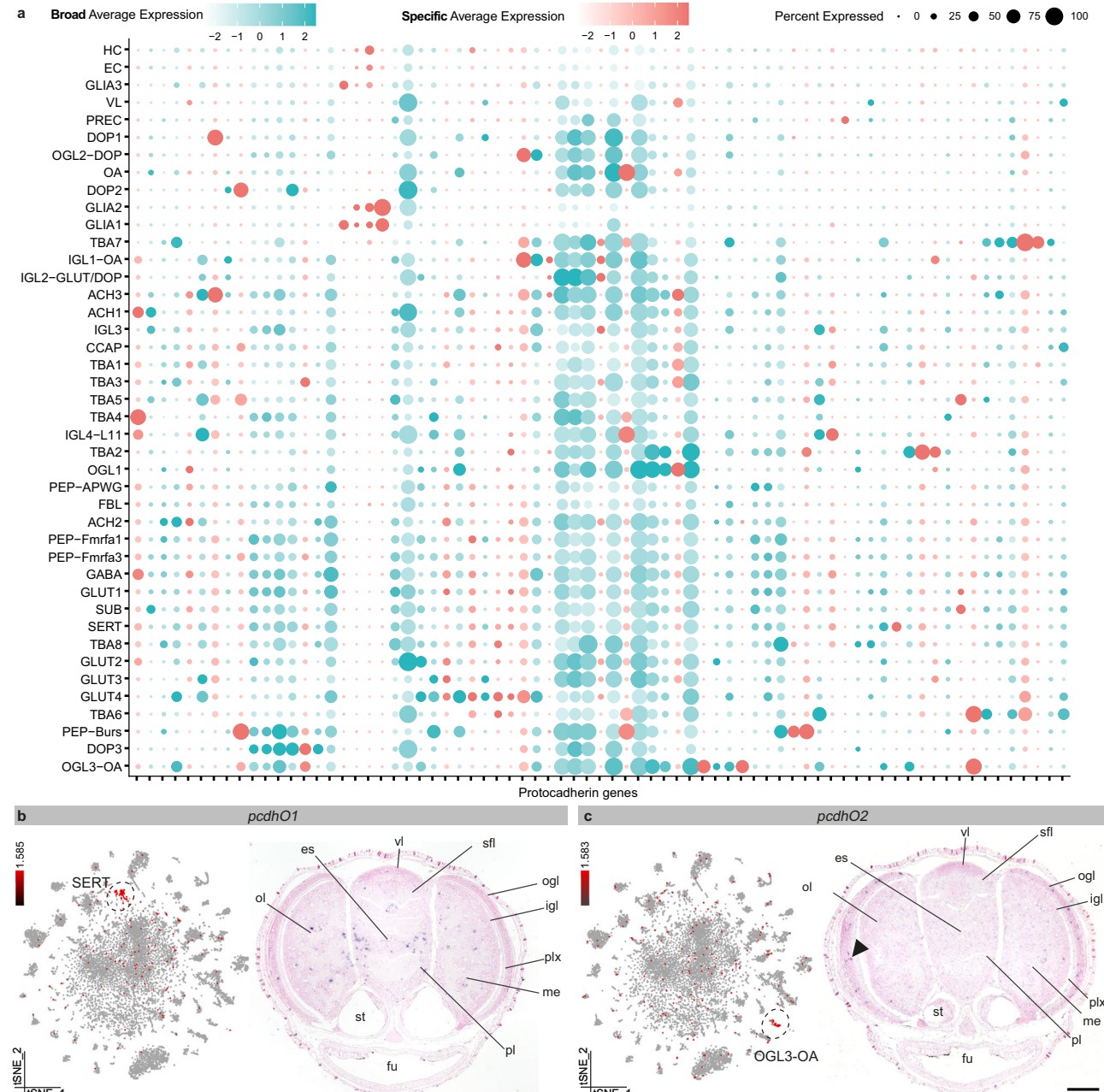

**Fig. 7 | Protocadherin gene family expansion underlies cellular diversification.** **a** Dot plot of highly variable protocadherin genes (PCDH). PCDH with a tissue specificity index above 0.85 are considered specific and color-coded in pink. PCDH with a tissue specificity index below 0.85 are broadly expressed and shown in blue. **b** Protocadherin O1 (*pcdhO1*) is expressed within serotonergic neurons (SERT).

**c** Protocadherin O2 (*pcdhO2*) is expressed within the octopaminergic neurons localized in the outer granular layer (OGL3-OA), a positive signal is indicated with a black arrow. Scale bars represent 100 μm. es esophagus, fu funnel, igl inner granular layer, me medulla, ogl outer granular layer, ol optic lobe, pl pedal lobe, plx plexiform layer, sfl superior frontal lobe, st statocysts, vl vertical lobe.

(data retrieved from PRJNA718058, PRJNA791920). We included both paralarval[25] and adult[32] Iso-Seq datasets of *O. vulgaris*. For each gene, the end of the longest isoform was considered the new 3′-end. Next, a full-length mRNA sequencing method—FLAM-seq—was used to locate mRNA cleavage sites in the genome (PRJNA791920)[32]. Cleavage sites located within 60,000 bp were assigned to the closest upstream genes. Finally, to account for the genes missing in the FLAM-seq dataset, published short-read RNA-seq datasets were used to extend the genes based on coverage (PRJNA547720)[98]. In brief, each gene was extended if there was sufficient continuous RNA-seq coverage (≥5 reads) downstream. A schematic depiction of the pipeline is available in Supplementary Fig. 1c. The resulting genome annotation is available in Supplementary Data 1. This approach resulted in a twofold decrease in the number of reads

mapping to intergenic regions (Supplementary Data 2). We manually curated the genome annotation for the PCDH gene family. Some read-through transcripts resulted in gene fusions and this was corrected by taking into account the number of protein domains. Transdecoder (https://github.com/TransDecoder/, v5.5.0) was used to identify the CDS. Functional annotation was performed by running BLAST + v2.7.1 against the SwissProt protein databases of *Drosophila melanogaster, Mus musculus*, and *O. bimaculoides* (with an e-value threshold of $10^{-5}$). In addition, EggNOG-mapper v2[99] was used to infer orthologies to bilaterian genes. The results are summarized in Supplementary Data 3. Gene ontology terms were also predicted by EggNOG, and we calculated the enriched gene ontology terms for certain clusters with the GSEApy package (v0.10.3).

## Animals

*O. vulgaris* embryos were obtained from the Instituto Español de Oceanografía (IEO, Tenerife, Spain). Embryos were then incubated until hatching in a closed system in the Laboratory of Developmental Neurobiology (KU Leuven), Belgium[23]. One day after hatching, larvae were sedated with 2% ethanol (in artificial seawater). Next, 30 brains were dissected on ice for single cells and 30 brains for single nuclei in L15-medium (Sigma) with additional salts (214 mM NaCl, 26 mM MgSO$_4$x7H$_2$O, 4.6 mM KCl, 2.3 mM NaHCO$_3$, 28 mM MgCl$_2$×6H$_2$O, 0.2 mM L-glutamine, 38 mM D-glucose, 10 mM CaCl$_2$×2H$_2$O, pH = 7.6). Statocysts and retinal tissues were removed as much as possible. All procedures involving hatchlings were approved by the ethical board on animal experimentation from KU Leuven (permit P080/2021), in compliance with Directive 2010/63/EU[100].

## Immunohistochemistry and in situ hybridization

One-day-old paralarvae were sedated as above and fixed overnight in 4% paraformaldehyde (PFA) in phosphate-buffered saline (PBS). Immunohistochemistry and in situ hybridization were performed as previously described[25]. Briefly, embryos were embedded in paraffin after progressive dehydration and sectioned with a paraffin microtome (Thermo Scientific, Microm HM360) to obtain 6-µm-thick transversal sections. For immunohistochemistry, we used a 1:300 dilution of monoclonal mouse anti-Acetylated alpha Tubulin (Sigma T6793, clone 6-11B-1, BATCH 0000108923) and polyclonal rabbit anti-phospho-histone H3 (Ser10) (Millipore 06-570, LOT3527703) as primary antibodies. Secondary antibodies Donkey anti-Mouse IgG (H + L) Antibody, Alexa Fluor 488 (Life Tech, Invitrogen, A-21202, LOT2266877), and Donkey anti-Rabbit IgG (H + L) Highly Cross-Adsorbed Antibody, Alexa Fluor 555 (Life Tech, Invitrogen, A-31572, LOT2286312) were also diluted 1:300. Colorimetric in situ hybridization was performed using DIG-labeled antisense probes and an automated platform (Ventana Discovery, Roche) with RiboMap fixation and BlueMap detection kits (Roche). Probes were designed to be between 500–1000 bp in length and were blasted against the *O. sinensis* genome to ensure specificity. The amount of probe used per slide (100–300 ng) and incubation with BCIP/NBT (6–9 h) was dependent on the target gene. Each probe was tested at least twice (different embryos and independent experiments). Primers and probe sequences are listed in Supplementary Data 6. Hybridization chain reaction (HCRv3.0) and imaging was performed as described before[25]. Briefly, probe sets were designed with the insitu_probe_generator[101] followed by automated blasting and formatting to minimize off-target hybridization with a custom script[102]. Probe sets were ordered for *Ov-glut*, *Ov-th*, *Ov-vacht*, *Ov-LOC118767670, Ov-apolpp*, and *Ov-gat1* from Integrated DNA Technologies, Inc (Supplementary Data 6). Amplifiers were ordered from Molecular Instruments, Inc (B1 Alexa Fluor-546, B2 Alexa Fluor 647, and B3 Alexa Fluor 488). To improve the signal-to-noise ratio, the probe concentrations were increased to 0.9 pmol. Imaging was done with a Leica DM6 upright microscope (IHC, colorimetric ISH), an Olympus confocal microscope Fluoview FV1000 or a Zeiss LSM900 (HCR).

## Single cell suspension

Paralarval brains were enzymatically dissociated by adding 20 µl of Collagenase/Dispase (100 mg/ml, Roche) to 500 µl L15-adapted medium (see above) and incubating for two hours at 25 °C, 500 rpm. Every 15 min, a P100 was used to pipet slowly up and down until the tissue was fully dissociated. After a 5 min centrifugation step (200×g, 4 °C), the supernatant was discarded and the pellet was resuspended in 1 ml of Mg-Ca-Free filtered seawater with 0.04% BSA (449 mM NaCl, 33 mM Na$_2$SO$_4$, 9 mM KCL, 2.15 mM NaHCO$_3$, 10 mM Tris-Cl pH 8.2, 2.5 mM EGTA, filter sterilized). The cells were pulled through a strainer (35 µm) by a brief spin, followed by a wash with 400 µl Ca-Mg-Free filtered seawater. Cells were centrifuged again for 5 min (200×g, 4 °C),

supernatant was removed, and the pellet was resuspended in 100 µl Ca-Mg-Free filtered seawater with 0.04% BSA. The cell viability and concentration were assessed by the LUNA-FL Dual Fluorescence Cell Counter (Logos Biosystems). We obtained a single-cell suspension with a multiplet cell percentage of 2.6%. The average cell size was 9.1 µm. The cell suspension was further diluted to reach appropriate cell counts, and a final viability of 84.9% was obtained before proceeding with 10X Genomics.

## Single nuclei extraction

The brains were immediately transferred to a Dounce homogenizer (Sigma) containing 0.5 ml of ice-cold homogenization buffer (HB) (320 mM Sucrose, 5 mM CaCl$_2$, 3 mM Mg(OAc)$_2$, 10 mM Tris 7.8, 0.1 mM EDTA, 0.1% IGEPAL CA-360, 0.1 mM Phenylmethylsulfonyl fluoride, 1 mM β-mercaptoethanol with 5 µl RNasin Plus). The tissue was incubated in the HB for 5 min before starting homogenization. The tissue was homogenized with 10 manual gentle strokes (pestle A) + 10 manual gentle strokes (pestle B). The tissue homogenate was filtered through a 70 µm cell mesh strainer. Leftover contents on the strainer were washed with an additional 0.5 ml HB buffer. The homogenized tissue was incubated in HB on ice for 5 min. Leftover contents on the strainer were washed with an additional 1.65 ml HB, which added to a final volume of 2.65 ml. The nuclei homogenate in the HB was mixed with 2.65 ml of Gradient Medium (GM) (5 mM CaCl$_2$, 50% Optiprep, 3 mM Mg(OAc)$_2$, 10 mM Tris 7.8, 0.1 mM Phenylmethylsulfonyl fluoride, 1 mM β-mercaptoethanol). 29% density cushion was prepared by dilution of Optiprep with Optiprep Diluent Medium (150 mM KCl, 30 mM MgCl$_2$, 60 mM Tris pH 8.0, 250 mM sucrose). The nuclei suspension in the HB + GM mix was layered over the 29% cushion and centrifuged in an SW41Ti rotor (Beckman Coulter) at 7700 rpm and 4 °C for 30 min. The supernatant was removed with a Pasteur pipette, and the removal of the lower supernatant was done with a P200. The nuclei pellet was resuspended in 50 µl Resuspension Buffer (PBS, 1% BSA) and transferred to a new tube. The resuspended nuclei were counted using a LUNA-FL Dual Fluorescence Cell Counter (Logos Biosystems).

## 10X Genomics

Library preparations for the sc/snRNA-seq experiments were performed using 10X Genomics Chromium Single Cell 3′ Kit, v3 chemistry (10X Genomics, Pleasanton, CA, USA). We aimed for a targeted cell recovery of 6000–10,000 cells/nuclei. Post cell count and QC, the samples were immediately loaded onto the Chromium Controller. Single cell or single nuclei RNA-seq libraries were prepared using manufacturers' recommendations (Single cell 3′ reagent kits v3.1 user guide; CG000204 Rev D), and the library quality was assessed using Qubit (Thermo Fisher) and Bioanalyzer (Agilent) at different checkpoints. With a targeted sequencing coverage of 25–50 K reads per cell, single-cell libraries were sequenced on Illumina's NovaSeq 6000 platform (VIB nucleomics core, KU Leuven) using paired-end sequencing workflow and with recommended 10X; v3.1 read parameters (28-8-0-91 cycles). A total of 202,402,758 reads were obtained for the nuclei and 247,457,191 reads for the cells.

## 10x Data preprocessing

All samples were processed with 10x Genomics Cell Ranger 5.0.1 for mapping, barcode assignment, and counting. Introns were retained and the parameter –expected cells was set at 8000 for both samples. Sequencing metrics for both samples can be found in Supplementary Data 2. The 3′-end extended genome annotation described above was used as a reference (Supplementary Data 1). This resulted in a raw dataset of 20,957 genes by 14,265 cells for the single cells and 21,073 genes by 8910 cells for the single nuclei. Filtering and subsetting steps were done in Seurat v3.2.3[103]. Nuclei and cells with too high (>4000) or

too low (<400 for nuclei, <800 for cells) gene counts were filtered out to exclude doublets and empty droplets. Cells with a higher percentage (>5) of mitochondrial RNA were regressed out since these were likely of low quality and possibly dying. Genes expressed in less than 10 cells were excluded. Highly variable genes were identified with the default VariableFeatures() function in Seurat (nfeatures = 3000). The SCTransform scaling method was used and data integration of the cells and nuclei was done following the recommended Seurat vignette. This resulted in a filtered integrated dataset of 17,961 genes by 17,081 cells. Subsequent data analysis was done with Seurat v4.0.4.

### Cluster annotation

The package scclusteval was used to assess optimal clustering parameters to obtain the highest number of stable clusters[35]. By resampling and repeated clustering, we used the mean Jaccard indices as a metric for stability. Reclustering according to these optimal parameters (dims = 150, k.param = 10, resolution = 2) resulted in the highest number of stable clusters. Cluster identities were transferred to the Seurat object. We used the package SCopeLoomR (https://github.com/aertslab/SCopeLoomR; v0.13.0) to generate the loom file, to facilitate data exploration in SCope. The expression levels of the genes in the SCope t-SNE plots are Log transformed and visualized with a scale bar. To visualize the expression levels of three genes in the CMY color scale, expression values were normalized with sctransfrom. We obtained a total number of 87 clusters, and for cluster annotation purposes, we filtered out all unstable clusters (<0.6 Jaccard index) and discarded the clusters that were not well defined (clusters 0,8,12,17,48,58). Cluster 3 and cluster 15 were merged into IGL2-GLUT/DOP. These two clusters largely overlapped and did not have many differentially expressed genes. We attributed this to a batch effect of the nuclei and cells. This resulted in a dataset of 42 robust clusters. Differentially expressed genes (Supplementary Data 5) were calculated for all clusters compared to all other clusters (min.pct = 0.25, logfc.threshold = 0.25). Cell-type annotation was based on the expression of vertebrate and invertebrate marker genes. Cell types were named based on their spatial localization and/or their neurotransmitter/neuropeptide phenotypes (Supplementary Data 4). The PrctCellExpringGene function was used to calculate the % of cells that expresses a certain gene (number of cells with raw counts >0). The differentiation state of all neuronal subtypes was assessed with iCytoTRACE[46]. The integrated dataset was subsetted for the nuclei and cells and iCytoTRACE was run on the raw count matrices (Supplementary Fig. 6). Bulk RNA-seq data from the adult nervous system of *O. vulgaris* was publicly available (https://www.ebi.ac.uk/biostudies/arrayexpress/studies/E-MTAB-3957?query=E-MTAB-3957)[49]. The top 100 differentially expressed genes between the optic lobes and the central brain were identified with DEseq2 and visualized with module scoring on the t-SNE plot (Supplementary Fig. 7). The central constellation was further analyzed by subclustering the data. The top 20 highly variable genes and transcription factors were identified and visualized in a dot plot (Supplementary Fig. 8c, d).

### Cross-species cell-type comparison

SAMap v0.1.6[60] was used to compare our data to scRNA-seq datasets of different species to gain more information about the identity and evolution of the octopus cell types. The octopus paralarval brain dataset was mapped to a mouse brain dataset[59] and to the adult fly brain[47]. Only alignment scores above 0.25 were considered to be of significance. Resulting annotations were visualized on the octopus t-SNE plot (Fig. 4) and listed in Supplementary Data 4.

### Cell-type tree construction

Transcription factors (TFs) were annotated with animalTFDB[104] and Possvm[105]. To infer cell-type relationships, a neighbor-joining tree was constructed based on the averaged expression values per cluster for all

genes as described previously[106]. Briefly, the R package Ape[107] was used to construct the cell-type tree (B = 10,000) and iTOL[108] for visualization. Bootstrap values between 50 and 100% were plotted on the tree branches with increasing dot size (Fig. 6a). Several gene sets were tested to construct the tree but using all the detected genes resulted in the highest bootstrap values.

### Transcription factors and cell-type specificity

Gene expression was averaged per cell type based on the SCT assay and TFs expressed in less than 20 cells were excluded. The tau value for all TFs was calculated using the tspex Python package (v0.6.2). TF family enrichment was calculated with GSEApy (v0.10.3) within the rank of tau (Fig. 6c, full results shown in Supplementary Data 7). The ComplexHeatmap R package was used for data visualization (Fig. 6d).

### Gene family enrichment analysis

Fisher's exact test was performed to calculate statistical enrichment for recently expanded gene families such as PCDH, C2H2- ZnF, and GPCR. Contingency tables were constructed and we then compared the number of genes belonging to a certain gene family to all other genes present in that cell type versus all other cell types. Only genes with an avg_logFC of above 0.25 or below −0.25 were considered for this analysis (75 PCDH, 141 C2H2-ZnF, and 130 GPCR). Gene lists are available as Supplementary Data 8. Fisher's exact tests for each cell type were followed by a Bonferroni correction for multiple testing with p.adjust() in R studio. Expression of octopus-specific genes was averaged per cell type based on the SCT assay and visualized on a scaled heatmap (Supplementary Fig. 21). The ComplexHeatmap R package was used for data visualization and significant enrichments were highlighted in red (Supplementary Fig. 21). The tau value was calculated for all PCDH with the tspex python package (v0.6.2) to analyze cell-type specificity.

### Statistics and reproducibility

We sequenced both the cellular and the nuclear transcriptomes and analyzed cell types present in both datasets. All probes for in situ hybridizations (colorimetric and HCR) were tested at least two times during different experiments (the exact number of replicates are detailed in Supplementary Data 6). Representative brain sections were chosen for each gene. Immunohistochemical stainings were repeated several times. Statistical analysis and data visualization was performed in R unless mentioned otherwise.

### Reporting summary

Further information on research design is available in the Nature Portfolio Reporting Summary linked to this article.

## Data availability

The scRNA-seq and snRNA-seq data generated in this study have been deposited in the Gene Expression Omnibus database under accession code GSE193622. The integrated and annotated datasets used in this study are also available online at https://scope.aertslab.org/#/Octopus_Brain/. SCope allows for easy simultaneous visualization of the expression of three genes while toggling between different embeddings. Marker gene lists can be downloaded here for different clusterings (Seurat clustering and the annotated clustering are also available in Supplementary Data 5). Different metrics can be visualized, such as the nCount, percent.mito, nFeature, and whether these originated from cells or nuclei (batch).

## Code availability

The R and python code that was used in this study is available on GitHub (https://github.com/SeuntjensLab/Styfhals_2022, https://github.com/rajewsky-lab/octopus_microRNAs/tree/main/gene_extension), and archived at Zenodo[109].

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

## Acknowledgements

The authors would like to thank Eduardo Almansa for his continuous support in providing us with octopus eggs and for sharing his expertise in octopus culture. Moreover, we want to thank Tania Aerts, Luca Masin, Mark Lassnig, and Ayana Rajagopal for critical discussions and Nikolai Hecker for his help with bioinformatic analysis. We would also like to acknowledge our master thesis students Swell Sieben for help during dissections and Sofia Maccuro and Dries Janssen for cloning. All computational analyses were performed at the Vlaams Supercomputer Center (VSC). This work was supported by KU Leuven (C14/21/065 to E.S., ID-N/20/007 to E.S. and S.A., and C14/18/092 to S.A.), Stazione Zoologica Anton Dohrn (G.P. and G.F.) and FWO (SB/1S19517N; A.D and FR/11D4120N; A.M.E.). R.S. was funded by a joint KU Leuven—Stazione Zoologica Anton Dohrn Ph.D. fellowship.

## Author contributions

R.S., S.P., A.M.E., and A.D. performed the experiments. R.S., G.Z., G.H., K.I.S., A.M.E., S.D.W., and E.S. analyzed and interpreted the data. R.S., S.A., and E.S. designed and G.P., N.R., G.F., S.A., and E.S. supervised the study. R.S. and E.S. wrote the original draft of the manuscript. All authors contributed to reviewing and editing.

## Competing interests

The authors declare no competing interests.
