## [Peer Review File · Nature Communications]

Cell type diversity in a developing octopus brainREVIEWER COMMENTS

Reviewer #1 (Remarks to the Author):

Styfals and colleagues present a manuscript leveraging new single-cell sequencing data to understand cell type diversity in the paralarval brain of *Octopus vulgaris*. As the *O. vulgaris* genome assembly is not well resolved, they employ an idiosyncratic method to map their reads onto the genome of *Octopus sinensis*, which has a chromosomal scale assembly. They produce Isoseq and FLAMseq to improve the annotation associated with their scRNAseq reads. They then sequence both single cells and nuclei from dissociated 1-day old hatchling octopus brains. Their analysis recovers 42 clusters, some better resolved than others, that correspond to cell types in the octopus brain, while it appears that the majority of cells reside in a central, unresolved cluster. The authors describe a select cell type populations using in situ hybridization.

In general, this manuscript would benefit from more explanation for the findings. As presented, this atlas requires the reader to be conversant in the many brain territories of the octopus: the introduction spans 114 lines, which is not sufficient to orient the reader. For example, what is the significance of the deep retina (discussed starting on line 359)? Why is a cholinergic cell type in the SUB interesting (line 356)? While some of these results are discussed at the end of the manuscript, combining the results and discussion will make this manuscript accessible to a wider audience.

While the authors distinguish 42 cell clusters, the majority of cells reside in a central constellation. While over-parsing data is of course inadvisable, much of the data presented here is unresolved. Did the authors try any of the other available methods for clustering/dimensionality reduction? Given the difficulty of mapping reads to the genome of another species (where more than half of the data is discarded from the outset), have the authors considered simply mapping the reads to their Isoseq/transcriptome assembly? Does treating the nuclei vs the cell transcriptomes separately result in different clustering?

From their description, the authors searched for the highest number of stable clusters. However, might this approach bias the results to retrieve the highest number of clusters, rather than trying to resolve the largest number of cells/nuclei? Employing and comparing multiple clustering methods might provide other, new metrics for interpreting the results.

The authors find glutamatergic, cholinergic, and dopaminergic cells make up most of the hatchling octopus brain. However, the multicolor HCR/in situ presented in Figure 2 is difficult to interpret: 1. The colors selected are difficult to resolve, and are not colorblind friendly. It is very difficult to differentiate the vglut signal relative to the th or vacht. 2. The individual colors should be presented to aid the reader in interpreting the results. 3. It also appears that there is very little co-expression between the vglut and the th in the section presented in 2d as suggested in the text, which I would expect to read out as yellow. If there is overlap, the authors should point it out clearly.

In the subsequent descriptions of cell types, the authors list a handful of genes that are expressed select cell type clusters, but they give little indication of what these genes are. For example, the authors identify a cholinergic cell type in the subesophageal mass (line 356). It would help the reader if (1) they point out the subesophageal mass in either of the figures they reference, (2) they tell us what the function of their marker genes are and how they were selected, and (3) more importantly, why is this result of interest or importance, other than to delineate a cell type? While I specifically point to this is early instance in the manuscript, the same questions could be asked of the description of the optic lobe cell types, vertical lobe cell types, etc. In general, the authors should carefully consider the genes they select as markers – for example, I would not expect that notch would be a specific cell type marker, and indeed, the authors use it as a marker for multiple cell types. Why are they pointing this out? How and why do they select the markers that they choose to discuss?

Why do the authors assume that the *vegf+* *vwf+* cells are hemocytes (line 498) rather than endothelial cells?

I am struck by the observation that the majority of cell types in the octopus brain seem to be molecularly distinct from those in the well characterized brains of mammals and flies. scRNAseq has the potential to reveal cell type diversity: could the authors also highlight unexpected cell types that are revealed as part of this analysis?

The authors investigate the expression of PCDH in these octopus brain cell types, but they point out that “these PCDH were not homologous to vertebrate PCDH” (line 529). What do the authors mean by this? How are they naming the PCDH – is PCDH15 related to the sequence in vertebrates, or is it just a randomly assigned number? In Fig 7a, the authors show a dotplot of “highly variable” PCDH expression – across this plot, many of the PCDH show a very similar pattern of expression: the middle 15% appear to be highly expressed across almost all the neuronal cell types, and many of the PCDH appear to have very similar dynamics across the different cell types. How was “highly variable” determined? While I can identify a handful that appear to be variable, could the authors use a different color or some other indication variable/divergent expression in the different cell types? What does this tell us about PCDH in the octopus brain? The authors suggest that unique combinations of PCDH could contribute to the complexity of the octopus brain. Is their hypothesis that individual PCDH are restricted to a specific cell type? Do the authors have examples of this from their data?

In such a fast-moving field, the authors should take care to limit claims of being the “first” (lines 547-549). There have been several Lophotrochozoans that have had single cell sequencing apart from *Platynereis* (see, for example, Swapna 2018; Fincher 2018; Sur 2021), including another mollusc (Salamanca-Diaz 2022).

Reviewer #2 (Remarks to the Author):

Styfals et al. characterize cellular diversity in the paralarval octopus brain using single cell and single nucleus RNA-seq. This work provides an important first step in describing the molecular diversity of neural cells in an understudied invertebrate species. It appears that only the most common or distinct cell types were captured, and that these types include neuronal and non-neuronal cells with varying spatial distributions, and a subset could be linked to cell types in the fly and mouse brain. This study also makes an important contribution by providing a better annotated octopus genome by leveraging additional transcriptomic datasets and manually curating expanded gene families.

The study would benefit from a more clearly annotated taxonomy of cell types that summarizes the various findings, including putative developmental origins based on marker gene expression, spatial distributions, and cross-species "homologies". The comparative analysis should be explained in more detail to show which functional classes of genes support the cell type similarities and what are the closest matches in fly and mouse for octopus cell types with no reported match. Are all of these novel types or are they related to types with more clear cross-species matches? The authors suggest that different TF networks regulate conserved sets of effector genes that may point to homologous cell types. This is quite different from conservation of cell types in the mammalian brain where conserved TFs drive expression of some conserved and many divergent effector genes. It would help to discuss more how profiling additional species would help differentiate convergent evolution from homology.

Specific comments:

- 42 cell types seems low given the number of cells that were profiled. How many were expected based on profiling other invertebrate species or morphological characterization of cells in octopus? Are fewer types detected because they are still acquiring a mature identity that is more distinct? Was an iterative clustering approach taken to select new sets of variable genes for subsets of cells to allow for further splitting?

- Was there any difference in cell type proportion estimates based on single nucleus and single cell data? Were any cell types more vulnerable to single cell dissociation and therefore single nucleus sequencing was beneficial?

- Fig. 1e, 2b - What is the interpretation of the unlabeled cells in the middle of the tSNE plot? Do they represent cell types with continuous variation or different cell states or rare types that cannot be separated? Please expand on this analysis by looking at QC metrics displayed on the tSNE to show that they are high quality. Also marker genes for neuronal activity that could potentially be conserved in invertebrates could highlight cell states. You can also look for genes with spatially enriched or gradient expression in portions of the central mass of neurons that would suggest cell subtypes may be identified with deeper sampling. Do you have ISH data to suggest the spatial distribution of cells in this large cluster?

- Fig. 2a - An additional analysis is to predict connectivity between types based on ligand/receptor expression, using CellPhoneDb or other similar tools.

- Fig. 4b - The results state that the glial marker signatures are conserved across the 3 species. Please show expression of these markers in fly and mouse homologous glial cell types.

- Fig. 4d, e - Do the three glial types have distinct anatomical distributions or morphology? Since two glial types map to mouse astrocytes that are present in the telencephalon or not, it would be important to know if these types show a similar segregation in the octopus nervous system that might point to shared functions or homologous structures.

- Fig. 4a - It would be helpful to include a visual summary of the comparisons of developmental origins that may be contributing cell type homologies, e.g. mouse microglia and octopus hemocytes.

- Fig. 6b - Do any of the TFs shown in the heatmap have known roles in octopus development or are related to the development of homologous types in other species? Does TF expression drive cell type homologies across species or is this due to other classes of genes?

- Fig. 7b,c - How similar are the transcripts for different protocadherin genes? How did you establish that the probes for pcdh15 and pcdh24 label only those transcripts?

- All of the ISH images would benefit from a corresponding schematic like Fig. S10 that shows the distribution of cells that are positive for the marker. It is difficult to see the ISH stain in many of the figures.

Reviewer #3 (Remarks to the Author):

Styfals et al manuscript presents a novel single cell dataset of the developing octopus brain. I found the study to be of interest and a valuable contribution to the community for the study of cell type evolution across species. This dataset can also be beneficial to further improve cross species analysis methods, which is still an under researched area that can shed light on human biology of disease. I commend the authors for the ingenious approach used for the genome annotation. The web application hosting the dataset is also a useful tool that will help reusing the data. I'm of the opinion that this work is a good fit for Nature Communications. However, before it can be published, I would like the authors to address a couple of comments which will improve the overall analysis quality and strengthen the manuscript as a whole.

Major points

The authors cite reference 6 as "extensive morphological characterization of the adult octopus brain". Can the authors provide a comparison between the classical literature reference and the data presented

in the manuscript? How many known cell types were found in the single cell data? How many are missing? Did the authors undersample and/or oversample some known functional groups?

10x data processing:

why do the authors have a different number of genes for nuclei and whole cell? Did the authors use a different reference genome?

Can the authors justify why they filter cells with both too high or too low gene counts and with more than 5% mito reads? Did the authors find any evidence between these variables and the presence of technical artifacts such as doublets or ambient RNA?

Following up from point 2b, the authors were not able to annotate the center of tSNE plots in Fig1 and subsequent figures. Did the authors look in detail at those cells and found any evidence of low quality metrics? It is not clear from the supplementary figure if a deep analysis was done and the rationale for keeping the non-annotated cells is also not discussed. Did the authors try to change the filtering parameters and observe any changes in the tSNE graph? Could a different embedding have helped finding whether the central region of the tSNE is mostly low quality transcriptomes or perhaps enrich the cell type annotations?

Data integration: the authors do not discuss the integration of whole cell and nuclei. The authors must at the very least discuss the advantages and limitations of each technology and how each and both combined empowered the analysis that are presented in the manuscript. I strongly disagree that “by using a dual approach in sequencing both cells and nuclei [they] have overcome any technical bias associated with each sequencing technology”. There is a considerable amount of literature discussing the comparison of the two technologies. Using both and ignoring the artifacts that each carries and that consequently will be present and likely affect the downstream analysis cannot be dismissed with such a sentence.

Minor points

Line 102, sentence starting with “Since the development of single-cell RNA sequencing technologies...” does not contribute to neither the previous nor the following sentence. Please rewrite.

Fig.3

consider changing the order of the panels by renumbering c->b, e->c, g->d, b->e, d->f and f->g

Improve the readability by clearly indicating in the figure in the right corresponds to the magnification of the rectangle

Fig.5

the staining of tmtc4 is not very clear, and in tSNE the expression of the gene is fairly scattered

Panel d - what do the different colors above the heatmap represent?

Can the authors elaborate about the possible orthologue for mki67 described in the non-neuronal cell types section?

Fig.6 - the authors should include the full results of the analysis summarized in Fig.6a as a supplementary table

Discussion - as the authors well document in introduction, cell type diversity in a mollusk has already been reported. Please rewrite the first sentence of the discussion section.

Fig.S3 - similar to minor point 3, the expression of GABA does not look specific

Fig.S9 - this figure is extremely hard to read, consider adding this to the web portal instead of having it as a supplementary figure

Cell type diversity in a developing octopus brain

Reply to the reviewers

We want to thank all the reviewers for their valuable feedback. We believe that due to their comments our manuscript improved significantly. Based on the comments of the reviewers, we have adapted 11 figures, and performed additional analyses leading to 14 novel supplementary figures, including an additional 3 figures and 2 tables for reviewers only. By including these new analyses, and the integration of the results and discussion, the text became significantly longer. To be in line with the formatting instructions, we removed text throughout the manuscript that was redundant or too speculative. We highlighted the main changes in **blue** in the main manuscript and have also added the respective line numbers in the answers to reviewers.

Reviewer #1 (Remarks to the Author):

Styfals and colleagues present a manuscript leveraging new single-cell sequencing data to understand cell type diversity in the paralarval brain of *Octopus vulgaris*. As the *O. vulgaris* genome assembly is not well resolved, they employ an idiosyncratic method to map their reads onto the genome of *Octopus sinensis*, which has a chromosomal scale assembly. They produce Iseq and FLAMseq to improve the annotation associated with their scRNAseq reads. They then sequence both single cells and nuclei from dissociated 1-day old hatchling octopus brains. Their analysis recovers 42 clusters, some better resolved than others, that correspond to cell types in the octopus brain, while it appears that the majority of cells reside in a central, unresolved cluster. The authors describe a select cell type populations using in situ hybridization.

- 1. In general, this manuscript would benefit from more explanation for the findings. As presented, this atlas requires the reader to be conversant in the many brain territories of the octopus: the introduction spans 114 lines, which is not sufficient to orient the reader. For example, what is the significance of the deep retina (discussed starting on line 359)? Why is a cholinergic cell type in the SUB interesting (line 356)? While some of these results are discussed at the end of the manuscript, combining the results and discussion will make this manuscript accessible to a wider audience.**

We thank the reviewer for this comment and agree that the brief introduction might not have been sufficient to understand all the results. We attempted to introduce some concepts better, such as the deep retina, and provided a brief summary in the introduction regarding the current knowledge of octopus neuronal cell types that might already be present at this life stage (lines 66-71).

Due to the overall paucity of knowledge on cell types, the information provided in the introduction reflects this limitation. We give an overview of the few studies describing neurotransmitters and neuropeptides at young/developmental life stages in these organisms (lines 76-80)

We also agree that combining results and the discussion would facilitate reading and implemented this change.

Finally, within the revised section 'Results and discussion' we also explained more clearly the relevance of the identified cell types such as the cholinergic cells in the sub (lines 416-422) and the deep retina (lines 436-437).

- 2. While the authors distinguish 42 cell clusters, the majority of cells reside in a central constellation. While over-parsing data is of course inadvisable, much of the data presented here is unresolved. Did the authors try any of the other available methods for clustering/dimensionality reduction?**

For dimensionality reduction, we only tried principal component analysis since this is recommended by Seurat¹ and performs well for datasets with a lot of variation. For clustering, we compared the Leiden and Louvain clustering algorithms and found that the separation of clusters improved with the latter. We found

that due to the lack of a priori knowledge regarding the expected number of cell types it was challenging to not over/under-cluster the dataset. Therefore, we opted to use *sclusteval* which gave us a reliable way to identify the optimal parameters for our dataset. We tried different clustering parameters (PC: 30, 50, 100, 150, Resolution: 0.6, 0.8, 1, 1.2, 1.4, 1.8, 2, 3, K-param: 8, 10, 12, 14, 16, 20, 30). We did this independently for the nuclei, cells and the integrated dataset. We identified the clustering parameters with the *highest number of stable clusters* (MOST_STABLE) and the clustering that resulted in the *highest percentage of cells assigned to stable clusters* (MOST_PERC) (see Table 1 hereunder).

For the integrated dataset we chose the clustering that had the highest number of stable clusters which resulted indeed in a large part of the data to be classified as ‘unstable’. We also annotated the clustering where the highest number of cells was assigned to stable clusters but found that this artificially grouped cell types. For data visualization, we used both t-SNE and UMAP. We observed the central constellation for all clustering settings (including only the cells and nuclei). Moreover, since we observe this constellation independently of the clustering method and technique, we presume this is a valid biological observation.

All reviewers had similar questions regarding the central constellation. We therefore further studied this enigmatic group of cell clusters in more detail. First, we compared our data with another invertebrate brain (*Drosophila*), which resulted in Fig. R1. In addition, we investigated their quality, whether they could be subclustered, and whether there were common features so we could hypothesize potential functions.

Table 1: Ideal clustering parameters for the different samples.

SAMPLE	PC	RES	K	TOTAL CLUSTER NUMBER
CELLS_MOST_STABLE	150	1.4	10	72
CELLS_MOST_PERC	150	0.6	16	35
NUCLEI_MOST_STABLE	50	3	10	67
NUCLEI_MOST_PERC	150	0.6	30	21
INT_MOST_STABLE	150	2	10	87
INT_MOST_PERC	50	0.6	30	27

a) Are these cells of low quality?

To investigate what is the biological difference between ‘central’ and ‘peripheral’ cells we grouped the cells based on their localization on the tSNE plot and investigated standard quality metrics, immediate early gene expression and transcription factors that were differentially expressed (Supplementary Fig. 5). The central constellation consists of high-quality cells (also see Supplementary Fig. 2d). This cluster has a high number of features, high counts and low mitochondrial gene expression (Supplementary Fig. 5b). The central constellation is clearly divided in two, based on the expression of *vglut* and *vacht* and therefore consists mainly of glutamatergic and cholinergic neurons (Fig. R1a, 2b). When we compare our dataset with another invertebrate brain atlas (fly) we observe a striking similarity (Fig. R1b). We added our findings to the manuscript (lines 356-365).

b) Are these immature cells?

We performed an additional analysis (iCytoTrace), which is based on the hypothesis that gene counts generally decrease with differentiation². The output of this analysis is visualized on the t-SNE plot in Supplementary Fig. 6. We found that the cells in the central constellation are predicted to be less differentiated. However, since there are peripheral clusters (SUB, SERT, IGL3) that also show high transcriptional diversity (i.e. less differentiated) we do not think that immaturity is the biological reason setting the central constellation apart. We also think it is unlikely that immature neurons already highly express markers of functional neurons such as *vacht* and *vglut*. We added our methods (lines 243-245) and findings (lines 365-370) to the revised manuscript.

c) *Do these cells represent rare cell types?*

When comparing our dataset with another published invertebrate brain dataset, the fly brain atlas, a similar central constellation was observed (Fig. R1a, b). In the fly brain, doubling the number of sequenced cells from 57k to 118k^{3,4} decreased the size of the central constellation and increased the number of distinct and separated clusters. The total number of cells present in the adult fly brain was estimated to be $199,380 \pm 3,400$ cells⁵. Even after sequencing around 60% of all cells, the central constellation was not resolved (Fig. R1b, d). Intriguingly, they did find that the central constellation mostly represents cell types that are localized within the central brain, while the more divergent clusters are cell types present in the optic lobe or very distinct cell types such as glial cells (Fig R1d)⁴. We also observed that the majority of stable (non-central-constellation) clusters we mapped spatially originated from the optic lobes. On the other hand, it is known that there are cell types present in very small numbers within the fly brain so this could also reflect the abundance of rare cell types, while in the optic lobe there might be more repetition and re-use of the same cell types. Whether the central constellation indeed represents more central brain cell types in octopus we could not distinguish at this stage. The issue was that 'marker' genes of the central constellation were not unique, but were often also expressed in the surrounding cell types. Therefore, we were unable to localize these cells within the brain via *in situ* hybridization. To circumvent this issue, we performed an additional analysis to identify whether the central constellation was more central brain-like or optic lobe-like by using publicly available bulk RNA-seq data of different regions of the adult octopus brain (Supplementary Fig. 7). We grouped the supra-esophageal mass and the sub-esophageal mass together as central brain and identified differentially expressed genes between the central brain and the optic lobes. We then used the module scoring function in Seurat to score all the cells based on the top 100 optic lobe genes and the top 100 central brain genes. We mostly identified cell types as 'optic lobe' that were already spatially mapped there (OGL2-DOP, IGL2-GLUT/DOP, IGL1-OA) and cell types we expected to map to the central brain (SERT, SUB, VL, GLUT4) indeed also mapped to their predicted location. Intriguingly, also populations whose locations are still unknown such as DOP1, DOP2, DOP3, OA showed high expression levels of adult optic lobe specific genes. Nevertheless, we found that the central constellation showed both expression of central brain and optic lobe specific genes. We therefore concluded that the central constellation had no specific spatial localization, but was a mixture of different rare cell types. Moreover, the presence of such big central constellations was also observed in recent single-cell studies on squid embryonic head⁶ and adult brain tissue⁷. We added our methods (lines 245-247) and findings (lines 384-391) to the revised manuscript.

d) *Is there heterogeneity within this cluster?*

To further understand what cell types might be hidden in the central constellation, we created a subset of the data containing only the central constellation and identified the most highly variable genes for these sub clusters. Still, even after subclustering, most of the data resided in a central constellation (Supplementary Fig. 8). We found that within this subset neuropeptides are overrepresented in the highly variable genes, and are especially marking the small clusters, suggesting that the central constellation consists of rare cell types that are neuropeptidergic (Supplementary Fig. 8c). We also identified transcription factors that are highly variable, including *rn/sqz*, a TF that regulates neuropeptidergic cellular identity (Supplementary Fig. 8d)⁸. Together, these analyses suggest that neuropeptidergic expression by rare cell types might be important in this young stage cephalopod brain. We added our methods (lines 247-249) and findings (lines 391-396) to the revised manuscript.

Fig. R1. Central constellation in *Octopus vulgaris*. **a** The central constellation consists mainly of glutamatergic (*vglut*) and cholinergic neurons (*vacht*). **b** A similar central constellation can be observed in the aging fly brain³. **c** Most of the neuronal cell types that surround the central constellation are localized in the optic lobes. **d** After sequencing 60% of all cells present in the fly brain, the central constellation remains and represents mainly central brain cell types.

3. Given the difficulty of mapping reads to the genome of another species (where more than half of the data is discarded from the outset), have the authors considered simply mapping the reads to their Isoseq/transcriptome assembly?

Although this is a valid comment, we have several reasons to believe our strategy was better. First, we did use an extensive database from both embryonic and adult *O. vulgaris* high quality IsoSeq data to annotate the *O. sinensis* genome, which significantly increased mapping statistics. Second, more importantly this strategy allowed us to minimize redundancy which is introduced when mapping to the transcriptome directly (Gene counts split over multiple transcripts etc.). Third, the genome of *O. sinensis* is chromosomal scale and *O. sinensis* is part of the same species complex as *O. vulgaris*. It was only recently discovered to be a different species⁹ and the genome was even wrongly submitted to NCBI as *O. vulgaris*. We therefore do not think a separate mapping to the transcriptome only would result in significantly better mapping statistics resolving the central constellation.

Still, in order to verify whether the species could be affecting the read-mapping, we performed an additional analysis. We used Cell Ranger v5.0.1 to map the scRNAseq reads for both the cells and the nuclei to the draft assembly of *O. vulgaris* using the same parameters as for *O. sinensis*¹⁰. We found that the percentage of reads that mapped to the genome was lower both for the cells and the nuclei (Table 1). This is likely due to the fragmented nature of the draft genome and its smaller size. As described before, we believe that the central constellation is biological and not a technical artefact related to the mapping.

Table 2: Percentage of reads mapped to the genome for the cells and nuclei.

SAMPLE	OCTOPUS SINENSIS	OCTOPUS VULGARIS
CELLS	88.4%	79.3%
NUCLEI	80%	69.8%

4. Does treating the nuclei vs the cell transcriptomes separately result in different clustering?

All reviewers raised a similar question. We therefore prepared an additional figure (Supplementary Fig. 3) to compare both datasets. Although most cell types have cells originating from both samples, we show in Supplementary Fig. 2c that the proportion of cells per cell type originating from cells versus nuclei is not always equal. Out of the 42 stable clusters, we observed 24 clusters with a slightly higher proportion of cells versus nuclei. **We found that the clustering and the number of stable clusters was similar for both samples.** We did see that through data integration we were able to distinguish more cell types and that obtained stable clusters were larger. We therefore decided to continue the annotation on the integrated dataset. We investigated the effect of integrating the cells and the nuclei by looking at the top 10 differentially expressed genes between the two datasets (Supplementary Fig. 3d). We found that in the nuclear RNA seq more lncRNA's were upregulated. In the cellular RNA seq we identified heat shock proteins and genes that might be linked to a stress response such as immediate early genes. We added these findings to the revised manuscript (lines 310-320).

5. From their description, the authors searched for the highest number of stable clusters. However, might this approach bias the results to retrieve the highest number of clusters, rather than trying to resolve the largest number of cells/nuclei? Employing and comparing multiple clustering methods might provide other, new metrics for interpreting the results.

As we mentioned above, we compared multiple clusterings. Our strategy was aimed to find the physiologically relevant cell types, even if these were rare. Indeed, we chose the clustering with the highest number of clusters since we opted for the resolution that was the most supported based on neurotransmitter or neuropeptide expression data. We found for instance that the predicted optimal resolution for the highest % of cells assigned to stable clusters was not high enough to accurately

separate the serotonergic neurons. We sought to avoid artificial grouping of cell types and found that the highest number of stable clusters supported our data the most.

- 6. The authors find glutamatergic, cholinergic, and dopaminergic cells make up most of the hatchling octopus brain. However, the multicolor HCR/in situ presented in Figure 2 is difficult to interpret: 1. The colors selected are difficult to resolve, and are not colorblind friendly. It is very difficult to differentiate the vglut signal relative to the th or vacht. 2. The individual colors should be presented to aid the reader in interpreting the results. 3. It also appears that there is very little co-expression between the vglut and the th in the section presented in 2d as suggested in the text, which I would expect to read out as yellow. If there is overlap, the authors should point it out clearly.**

We agree that Fig. 2 was difficult to interpret due to low signal intensity. We have optimized our protocol by increasing the probe concentration x3, repeated this *in situ* hybridization and have created two additional supplementary figures (Supplementary Fig. 9-10). We changed the RGB colors on the t-SNE to a more color-blind friendly CMY color scale, and provide the separate channels for the HCR in grey scale. In our new Fig. 2e and Supplementary Fig. 10g-j, co-expression is clearly visible as white in the inner granular layer of the optic lobe.

- 7. In the subsequent descriptions of cell types, the authors list a handful of genes that are expressed select cell type clusters, but they give little indication of what these genes are. For example, the authors identify a cholinergic cell type in the subesophageal mass (line 356). It would help the reader if (1) they point out the subesophageal mass in either of the figures they reference, (2) they tell us what the function of their marker genes are and how they were selected, and (3) more importantly, why is this result of interest or importance, other than to delineate a cell type? While I specifically point to this is early instance in the manuscript, the same questions could be asked of the description of the optic lobe cell types, vertical lobe cell types, etc. In general, the authors should carefully consider the genes they select as markers – for example, I would not expect that notch would be a specific cell type marker, and indeed, the authors use it as a marker for multiple cell types. Why are they pointing this out? How and why do they select the markers that they choose to discuss?**

We agree that the description would benefit from more explanation. We have accommodated several changes based on this comment:

- (1) To improve the readability, we annotated the subesophageal mass in Fig. 2c and Supplementary Fig. 11.
- (2) In the materials and method section we adjusted the statement on how we identified marker genes; lines 237-239.
- (3) Throughout the manuscript we have added more information for the marker genes when this was relevant and have written the gene names in full. Indeed, in order to localize cell types within the brain we sought to use markers that were uniquely expressed only in that cell type. However, we did aim to use markers which have known functions wherever this was possible.

We added this information to the manuscript, for example see lines 436-437, 442-443 and 447-448.

We observed *notch* expression in the endothelial cells, hemocytes and in all glial subtypes similar to N expression in *Drosophila*. However, we agree that *notch* often has context-specific roles¹¹ and therefore we removed it from the list of 'conserved markers' of glia and endothelial cells throughout the manuscript.

8. Why do the authors assume that the vegf+ vwf+ cells are hemocytes (line 498) rather than endothelial cells?

Gene LOC115222969 was annotated as 'vwf' based on the presence of a von willebrand factor domain. We verified the annotation using blast analysis and decided to rename this gene to the more accurate abbreviation: *svep1*, 'sushi, von Willebrand factor type A, EGF and pentraxin domain-containing protein 1'. A bulk RNA-seq study performed on octopus hemocytes showed that they express both *vegf* and *vwf*¹² and a study on squid hemocytes found high expression of *svep1*¹³. Conversely, the endothelial cells highly express VegfR¹⁴.

9. I am struck by the observation that the majority of cell types in the octopus brain seem to be molecularly distinct from those in the well characterized brains of mammals and flies. scRNAseq has the potential to reveal cell type diversity: could the authors also highlight unexpected cell types that are revealed as part of this analysis?

If we compare our data with a well characterized invertebrate model species; *Drosophila melanogaster*, **between 87-151 clusters** were identified in the initial paper, of which the authors were able to annotate 37³. Regarding the morphological characterization of cells in the octopus brain, most of the work has been done on adults and indeed, it is likely that **the number of cell types might still increase** until adulthood, because a massive brain expansion is taking place between hatching and adult life (1000-fold increase in cell number) and the life-style of the animal changes drastically. Young described the anatomy of the adult nervous system of *Octopus vulgaris* mostly based on histological observations and Golgi stains¹⁵. This resource also includes a description of the cells present in the different lobes of the octopus brain¹⁶. Although on occasion the morphology of particular cell types is mentioned (e.g., amacrine cells), a detailed overview is missing. However, we can use the nuclear size as a metric for the number of cellular phenotypes¹⁶. Young classified the cells based on their location and created 5 bins for nuclear size (<5 um, 5-10 um, 10-15 um, 15-20 um, 20-25 um). A total of **116 different cellular phenotypes** (number of lobes x number of bins observed in each lobe) can be distinguished when making the following assumptions: 1) the chosen bins are accurate to discriminate cell types and 2) different brain territories or lobes possess different cell types. This number can therefore be an overestimation, since we show here that several cell types are distributed in different brain regions (e.g. serotonergic cells) or an underestimation since similar nuclear size does not necessarily result in a single cell type. We can suppose that the number of cell types present at this life stage indeed is much higher than 42, and likely ranges between 100-150. Since our analysis yielded 87 clusters, of which most seemed homogenous clusters based on marker gene expression, we therefore did not attempt to sub-cluster the data further. We did subset and re-cluster the central constellation using several parameters and although this did result in a lot of smaller clusters, the majority of cells were still unresolved (Supplementary Fig. 8).

Since there was not much a priori knowledge related to cell type diversity within the larval octopus brain, we also had no reference list of 'expected cell types'. Only a few studies regarding neurotransmitter and neuropeptide expression in developing cephalopod brains were published. We added this to the introduction after our literature review (lines 76-79). We used these data as well to sum up the most expected findings (lines 339-345 and 353-355).

Conversely, it was even less trivial to highlight 'unexpected cell types'. The last common ancestor between octopus and flies and humans lived more than 500 million years ago, therefore we do not consider it surprising that we could not identify more similarities between these species. Octopus neurons do exhibit general neuronal characteristics such as common neurotransmitters and neuropeptides. However, the diversification of neuronal subtypes likely happened later on during evolution which means that neuronal subtypes (and even glial subtypes) cannot be compared one-to-one across these distantly related species. All the putative homologies are listed in Supplementary Data 4. Although one could consider that cell types that did not have any match could be considered as molecularly distinct, this could also be due to the orthology annotations. Therefore, we opted for another method to identify

novel/unexpected cell types by looking at enrichments of octopus specific genes (Fig. 7). More data needs to be collected and compared between closely and more distantly related species to really identify octopus innovations. We believe there is much more to be discovered soon, and we hope this dataset can benefit the field even at this stage. However, here we summed up the most unexpected cell types;

a) A limited amount of precursor cells: Although the octopus paralarva is free-swimming at this developmental life stage and shows interesting innate behaviors such as predation and positive phototaxis, we presumed that there would be clear populations of progenitor cells connected to immature neurons and more mature neurons in this 'early life stage'. An unexpected finding was that we identified only a small population of precursor cells which we believe are the remnants of the lateral lip, the transient embryonic neurogenic niche in octopus, and no clear sign of neurogenesis or a neurogenic population. We believe this suggests that the recently hatched paralarvae are transitioning from an embryonic growth phase to paralarval life and are not generating new neurons at this stage.

b) A large diversity of mature optic lobe cell types: We looked at the spatial localization of several cell types within the octopus brain and for the most part we identified optic lobe cell types and based on our iCytoTRACE analysis these cell types were often considered to be more mature. This suggests that the optic lobe is more mature compared to other brain regions at this point and already exhibits a surprising amount of cell diversity (lines 455-457).

c) Many rare peptidergic neurons: The central constellation was an unexpected finding. Whether this is a trait of centralized invertebrate brains (*Drosophila*, *Octopus*), we do not know. We hypothesized above that the central cluster likely represents rare cell types that seem to be peptidergic. Besides those, we also identified two cell types (CCAP and PEP-burs) that express peptides involved in ecdysis in arthropods. We added this to the manuscript in lines 412-414.

10. The authors investigate the expression of PCDH in these octopus brain cell types, but they point out that “these PCDH were not homologous to vertebrate PCDH” (line 529). What do the authors mean by this? How are they naming the PCDH – is PCDH15 related to the sequence in vertebrates, or is it just a randomly assigned number?

Indeed, the expansion of PCDH in octopodiform cephalopods happened independently from that in decapodiform cephalopods, and also independently from vertebrate PCDH. Yet these molecules are true PCDH as they have 6 or 7 extracellular cadherin domains (ECD), a transmembrane domain (TM) and an intracellular domain, and the full ECD and TM and part of the cytoplasmic domain are encoded by a single first exon. Cephalopod PCDH are also organized in clusters, yet within each cluster, all PCDH genes encode a full PCDH, in contrast to vertebrate clustered PCDH where distinct ECD are spliced onto a shared intracellular domain¹⁷. In *Octopus bimaculoides* and in other cephalopods, the PCDH gene expansion is located fully on one chromosome¹⁸. Nomenclature should ideally follow the cluster order and gene order on the chromosome, similar to the way nomenclature was given to the clustered PCDH in vertebrates. At this point however, we are not able yet to assign numbers to PCDH in this manner, as the *Octopus vulgaris* and *Octopus sinensis* genome does not have full chromosome resolution and protocadherin genes are still spread across multiple scaffolds. At the onset of this study, we therefore simply numbered the PCDH genes. We do acknowledge that such a numbering has generated unnecessary confusion with vertebrate PCDH. We therefore renamed *pcdh15* to *pcdhO1* and *pcdh24* to *pcdhO2*.

11. In Fig 7a, the authors show a dotplot of “highly variable” PCDH expression – across this plot, many of the PCDH show a very similar pattern of expression: the middle 15% appear to be highly expressed across almost all the neuronal cell types, and many of the PCDH appear to have very similar dynamics across the different cell types. How was “highly variable” determined? While I can identify a handful that appear to be variable, could the authors use a different color or some other indication variable/divergent expression in the different cell types? What does this tell us about PCDH in the octopus brain? The authors suggest that unique combinations of PCDH could contribute to the complexity of the

octopus brain. Is their hypothesis that individual PCDH are restricted to a specific cell type? Do the authors have examples of this from their data?

Triggered by the many questions in this comment, we performed a number of additional analyses to understand how PCDH are expressed on the cell-type and individual cell level. We prepared a supplementary figure illustrating the expression of all PCDH (not only the highly variable PCDH) (Supplementary Fig. 22). Based on all data we can make the following conclusions; a) in Supplementary Fig. 22a we show that some PCDH are uniquely expressed in certain cell types, while others are more broadly expressed. b) non-neuronal cell types, such as EC, GLIA, and HC, have lower total PCDH expression levels (Supplementary Fig. 22b). To understand how PCDH might contribute to the complexity of the octopus brain we grouped the PCDH based on their genomic location. Several PCDH were located on Unplaced Scaffolds, and therefore a comprehensive annotation cannot yet be generated. However, we identified 159 'clustered' (located on the same chromosome) and 13 'non-clustered' PCDH (located on different scaffolds). We could not identify any clear differences in expression based on their genomic location (Supplementary Fig. 22a-b). Looking at raw PCDH counts of individual cells we find that on average each cell expresses 19 different PCDH genes (Supplementary Fig. 22c). If we look only at neuronal cells this number is higher (21 PCDH) than in non-neuronal cells (13 PCDH). We can therefore deduce that PCDH might play a role in non-neuronal cells but that they are probably more important in neurons. The observation that so many different PCDH are co-expressed in the same cells is in line with their role as a neuronal barcode that could be used for cell-cell recognition¹⁹. We do hypothesize that unique combinations of PCDH might contribute to the complexity of the octopus brain. We have several examples in our data that particular PCDH are restricted to a specific cell type; of which we show two in Fig 7b, c, but many other examples are present in Fig. 7a and Supplementary Fig. 22a. For example, it seems that there are several PCDH that are highly and exclusively expressed in non-neuronal cells (GLIA, HC, EC). We added our findings to the manuscript (lines 731-739).

In addition, in the materials and method section we added the following sentence; "*Highly variable genes were identified with the default VariableFeatures() function in Seurat (nfeatures = 3000).*" In order to indicate which PCDH are broadly expressed versus more specifically expressed we ordered these genes based on their tissue-specificity index tau (calculated in the same way as for the transcription factors). We replotted the expression values in a dot plot and all PCDH with a tau value above 0.85 were color-coded in pink and PCDH with a tau value below 0.85 were shown in blue (Fig. 7).

12. In such a fast-moving field, the authors should take care to limit claims of being the "first" (lines 547-549). There have been several Lophotrochozoans that have had single cell sequencing apart from Platynereis (see, for example, Swapna 2018; Fincher 2018; Sur 2021), including another mollusc (Salamanca-Diaz 2022).

Indeed, while this paper was in revision, three more preprints describing cephalopod single-cell data have come out. We have deleted these sentences. We thank the reviewer (also a comment from reviewer #3) for this suggestion.

Reviewer #2 (Remarks to the Author)

General comments

Styfals et al. characterize cellular diversity in the paralarval octopus brain using single cell and single nucleus RNA-seq. This work provides an important first step in describing the molecular diversity of neural cells in an understudied invertebrate species. It appears that only the most common or distinct cell types were captured, and that these types include neuronal and non-neuronal cells with varying spatial distributions, and a subset could be linked to cell types in the fly and mouse brain. This study also makes an important contribution by providing a better annotated octopus genome by leveraging additional transcriptomic datasets and manually curating expanded gene families.

The study would benefit from a more clearly annotated taxonomy of cell types that summarizes the various findings, including putative developmental origins based on marker gene expression, spatial distributions, and cross-species "homologies". The comparative analysis should be explained in more detail to show which functional classes of genes support the cell type similarities and what are the closest matches in fly and mouse for octopus cell types with no reported match. Are all of these novel types or are they related to types with more clear cross-species matches? The authors suggest that different TF networks regulate conserved sets of effector genes that may point to homologous cell types. This is quite different from conservation of cell types in the mammalian brain where conserved TFs drive expression of some conserved and many divergent effector genes. It would help to discuss more how profiling additional species would help differentiate convergent evolution from homology.

We think this comment is a very valid one, and we tried to implement the changes asked for. We reasoned that, in order to better tease apart convergence from homology, we needed to analyze in more depth the relationships between cell types and which transcription factors are specifying cell types. We constructed a neighbor-joining tree with 10,000 Bootstrap iterations, which is considered more robust than the tree originally shown in Fig. 1. The resulting taxonomy of cell types is presented in the revised paper (Fig. 6). We started the analysis by first improving our existing TF annotation with animalTFDB by using Possvm²⁰ to accurately annotate reference orthologues. By investigating which transcription factors were shared amongst the different branches of the tree (i.e. between closely related cell types) we could infer which transcriptional regulators were important in the octopus brain. We identified the major TFs (in Fig. 6a-b) which are important for cell type specification within the octopus brain. Intriguingly, in our taxonomy, more supported branches seemed to contain cell types that were spatially close together in the brain, suggesting that in different regions, cell diversification might have happened from an ancestral cell type. See methods lines 259-264. Our findings are described in lines 665-686.

We hope that fig. 6a provides a clearer taxonomy of cell types and how they are related. We also prepared additional Supplementary Figures 13, 16 and 17 to show which genes and TF underlie the putative cell type homologies.

We also added a statement to our conclusion regarding future work on cell type evolution (lines 761-762).

We will discuss the rest of the comments in our point-by-point response below.

Specific comments:

- 1. 42 cell types seems low given the number of cells that were profiled. How many were expected based on profiling other invertebrate species or morphological characterization of cells in octopus? Are fewer types detected because they are still acquiring a mature identity that is more distinct? Was an iterative clustering approach taken to select new sets of variable genes for subsets of cells to allow for further splitting?**

Indeed, we do not presume to have identified all the cell types in the octopus brain. Based on our initial analysis we obtained 87 distinct clusters of which 42 were considered stable (full list in Supplementary Data 4). We also added this information to our results (line 326).

If we compare our data with a well characterized invertebrate model species, *Drosophila melanogaster*, **between 87-151 clusters** were identified in the initial paper, of which the authors were able to annotate 37³. Regarding the morphological characterization of cells in the octopus brain, most of the work has been done on adults. Young described the anatomy of the adult nervous system of *Octopus vulgaris* mostly based on histological observations and Golgi stains¹⁵. This resource also includes a description of the cells present in the different lobes of the octopus brain¹⁶. Although on occasion, the morphology of particular cell types is mentioned (e.g. amacrine cells), a detailed overview is missing. However, we can use the nuclear size as a metric for the number of cellular phenotypes¹⁶. Young classified the cells based on their location and created 5 bins for nuclear size (<5 um, 5-10 um, 10-15 um, 15-20 um, 20-25 um). A total of **116 different cellular phenotypes** (number of lobes x number of bins observed in each lobe) can be distinguished when making the following assumptions: 1) the chosen bins are accurate to discriminate cell types, and 2) different brain territories or lobes possess different cell types. This number can therefore be an overestimation, since we show here that several cell types are distributed in different brain regions (e.g. serotonergic cells) or an underestimation since similar nuclear size does not necessarily result in a single cell type. We can suppose that the number of cell types present at this life stage indeed is much higher than 42, and likely ranges between 100-150. Since our analysis yielded 87 clusters, of which most seemed homogenous clusters based on marker gene expression, we therefore did not attempt to sub-cluster the data further. We did subset and re-cluster the central constellation using several parameters and although this did result in a lot of smaller clusters, the majority of cells still resided in the middle (Supplementary Fig. 8).

It is likely that **the number of cell types will still increase** until adulthood, because a massive brain expansion is taking place between hatching and adult life (1000-fold increase in cell number) and the life-style of the animal changes drastically. To investigate the differentiation state of the cells we performed an additional analysis which is based on the transcriptional diversity (Supplementary Fig. 6). We found that most cells were not fully differentiated yet (see below).

To make it clear that more cell types are likely present, we added lines 755-757 to the conclusion.

Was there any difference in cell type proportion estimates based on single nucleus and single cell data? Were any cell types more vulnerable to single cell dissociation and therefore single nucleus sequencing was beneficial?

All reviewers raised a similar question. We therefore prepared an additional figure (Supplementary Fig. 3) to highlight the differences between the two datasets. Although most cell types have cells originating from both samples, we show in Supplementary Fig. 2c that the proportion of cells per cell type originating from cells versus nuclei is not always equal. Out of the 42 stable clusters, we observed 24 clusters with a slightly higher proportion of cells versus nuclei. We found that the clustering and the number of stable clusters was similar for both samples. We did see that through data integration we were able to distinguish more cell types and that obtained stable clusters were larger. We therefore decided to continue the annotation on the integrated dataset. We investigated the effect of integrating the cells and the nuclei by looking at the top 10 differentially expressed genes between the two datasets (Supplementary Fig. 3d). We found that in the nuclear RNA seq more lncRNA's were upregulated. In the cellular RNA seq we identified heat shock proteins and genes that might be linked to a stress response such as immediate early genes. We added these findings to the revised manuscript (lines 310-320).

2. Fig. 1e, 2b - What is the interpretation of the unlabeled cells in the middle of the tSNE plot? Do they represent cell types with continuous variation or different cell states or rare types that cannot be separated? Please expand on this analysis by looking at QC metrics displayed on the tSNE to show that they are high quality. Also marker genes for neuronal activity that could potentially be conserved in invertebrates could highlight cell states. You can also look for genes with spatially enriched or gradient expression in portions of the central mass of neurons that would suggest cell subtypes may be identified with deeper sampling. Do you have ISH data to suggest the spatial distribution of cells in this large cluster?

Fig. R1. Central constellation in *Octopus vulgaris*. **a** The central constellation consists mainly of glutamatergic (*vglut*) and cholinergic neurons (*vacht*). **b** A similar central constellation can be observed in the aging fly brain³. **c** Most of the neuronal cell types that surround the central constellation are localized in the optic lobes. **d** After sequencing 60% of all cells present in the fly brain, the central constellation remains and represents mainly central brain cell types.

All reviewers had similar questions regarding the central constellation. We therefore further studied this enigmatic group of cell clusters in more detail. First, we compared our data with another invertebrate brain (*Drosophila*), which resulted in Fig. R1. In addition, we investigated their quality, whether they could be subclustered and whether there were common features so we could hypothesize potential functions.

Table 1: Ideal clustering parameters for the different samples.

SAMPLE	PC	RES	K	TOTAL CLUSTER NUMBER
CELLS_MOST_STABLE	150	1.4	10	72
CELLS_MOST_PERC	150	0.6	16	35
NUCLEI_MOST_STABLE	50	3	10	67
NUCLEI_MOST_PERC	150	0.6	30	21
INT_MOST_STABLE	150	2	10	87
INT_MOST_PERC	50	0.6	30	27

a) *Are these cells of low quality?*

To investigate what is the biological difference between ‘central’ and ‘peripheral’ cells we grouped the cells based on their localization on the tSNE plot and investigated standard quality metrics, immediate early gene expression and transcription factors that were differentially expressed (Supplementary Fig. 5). The central constellation consists of high-quality cells (also see Supplementary Fig. 2d). This cluster has a high number of features, high counts and low mitochondrial gene expression (Supplementary Fig. 5b). The central constellation is clearly divided in two based on the expression of *vglut* and *vacht* and therefore consists mainly of glutamatergic and cholinergic neurons (Fig. R1a, 2b). When we compare our dataset with another invertebrate brain atlas (fly) we observe a striking similarity (Fig. R1b). We added our findings to the manuscript (lines 356-365)

b) *Are these immature cells?*

We performed an additional analysis (iCytoTrace), which is based on the hypothesis that gene counts generally decrease with differentiation². The output of this analysis is visualized on the t-SNE plot in Supplementary Fig. 6. We found that the cells in the central constellation are predicted to be less differentiated. However, since there are peripheral clusters (SUB, SERT, IGL3) that also show high transcriptional diversity (i.e. less differentiated) we do not think that immaturity is the biological reason setting the central constellation apart. We also think it is unlikely that immature neurons already highly express markers of functional neurons such as *vacht* and *vglut*. We added our methods (lines 243-245) and findings (lines 365-370) to the revised manuscript.

c) *Do these cells represent rare cell types?*

When comparing our dataset with another published invertebrate brain dataset; the fly brain atlas, a similar central constellation was observed (Fig. R1a, b). In the fly brain, doubling the number of sequenced cells from 57k to 118k^{3,4}, decreased the size of the central constellation and increased the number of distinct and separated clusters. The total number of cells present in the adult fly brain was estimated to be 199,380 ± 3,400 cells⁵. Even after sequencing around 60% of all cells, the central constellation was not resolved (Fig. R1b, d). Intriguingly, they did find that the central constellation mostly represents cell types that are localized within the central brain, while the more divergent clusters are cell types present in the optic lobe or very distinct cell types such as glial cells (Fig R1d)⁴. We also observed that the majority of stable (non-central-constellation) clusters we mapped spatially originated from the optic lobes. On the other hand, it is known that there are cell types present in very small numbers within the fly brain so this could also reflect the abundance of rare cell types, while in the optic lobe there might be more repetition and re-use of the same cell types. Whether the central constellation indeed represents more central brain cell types in octopus we could not distinguish at this stage. The issue was that ‘marker’ genes of the central constellation were not unique, but were often also expressed in the surrounding cell

types. Therefore, we were unable to localize these cells within the brain via *in situ* hybridization. To circumvent this issue, we performed an additional analysis to identify whether the central constellation was more central brain-like or optic lobe-like by using publicly available bulk RNA-seq data of different regions of the adult octopus brain (Supplementary Fig. 7). We grouped the supra-esophageal mass and the sub-esophageal mass together as central brain and identified differentially expressed genes between the central brain and the optic lobes. We then used the module scoring function in Seurat to score all the cells based on the top 100 optic lobe genes and the top 100 central brain genes. We mostly identified cell types as 'optic lobe' that were already spatially mapped there (OGL2-DOP, IGL2-GLUT/DOP, IGL1-OA) and cell types we expected to map to the central brain (SERT, SUB, VL, GLUT4) indeed also mapped to their predicted location. Intriguingly, also populations whose locations are still unknown such as DOP1, DOP2, DOP3, OA showed high expression levels of adult optic lobe specific genes. Nevertheless, we found that the central constellation showed both expression of central brain and optic lobe specific genes. We therefore concluded that the central constellation had no specific spatial localization, but was a mixture of different rare cell types. Moreover, the presence of such big central constellations was also observed in recent single-cell studies on squid embryonic head⁶ and adult brain tissue⁷. We added our methods (lines 245-247) and findings (lines 384-391) to the revised manuscript.

d) *Is there heterogeneity within this cluster?*

To further understand what cell types might be hidden in the central constellation, we created a subset of the data containing only the central constellation and identified the most highly variable genes for these sub clusters. Still, even after subclustering, most of the data resided in a central constellation (Supplementary Fig. 8). We found that within this subset neuropeptides are overrepresented in the highly variable genes, and are especially marking the small clusters, suggesting that the central constellation consists out of rare cell types that are neuropeptidergic (Supplementary Fig. 8c). We also identified transcription factors that are highly variable, including *rn/sqz*, a TF that regulates neuropeptidergic cellular identity (Supplementary Fig. 8d)⁸. Together, these analyses suggest that neuropeptidergic expression by rare cell types might be important in this young stage cephalopod brain. We added our methods (lines 247-249) and findings (lines 391-396) to the revised manuscript.

3. Fig. 2a - An additional analysis is to predict connectivity between types based on ligand/receptor expression, using CellPhoneDb or other similar tools.

We thank the reviewer for this suggestion, but we would like to argue why at this stage these tools might not yet reliably work. First of all, we would need validation that ligands and receptors predicted in our annotation are true homologs, and operate functionally in similar ways, otherwise any information deduced from these tools might be invalid. Indeed, we already found in a separate study on ligand-gated ion channels that, in some cases, the predicted ligand does not always turn out to be the real activating ligand. Receptor annotation thus would need functional validation to make these tools really reliable. In addition, we know of other groups that are working on elucidating neuronal connectivity in cephalopod brains through electron microscopy which will be more informative.

Instead of analyzing cell type connectivity, we thought it would be interesting to highlight which cell types are more evolutionary related. Therefore, we constructed a phylogenetic tree and analyzed which transcription factors are important regulators (see above).

4. Fig. 4b - The results state that the glial marker signatures are conserved across the 3 species. Please show expression of these markers in fly and mouse homologous glial cell types.

The heatmap in the original Fig. 4b represented "*the top 15 genes (filtered on specificity and fold change) from the mapping between octopus glia 1, fly ensheathing glia and non-telencephalic astrocytes in the mouse brain.*" SAMap is however not restricted to one-to-one orthologues since paralogs might be more functionally similar across species and there are only very few one-to-one orthologues across such a large evolutionary scale. This means we could not simply retrieve expression of these top-15 genes across the different species, because these species might utilize different paralogs. Based on this

comment from the reviewer we reasoned that it might be more feasible and relevant to search for the true one-to-one orthologues as part of the common signature. We thus decided to revise this gene list and instead of taking the highest expressed genes we selected the common putative one-to-one gene orthologues across these three species. We therefore filtered the 196 common octopus genes that came out from both the fly and mouse mapping based on reciprocal blast. We identified 20 octopus genes from this list that had a one-to-one orthologue in both the mouse and the fly. We represented the expression of this gene set on a heatmap in fig 4b, and show the expression of these markers in the fly and mouse in Supplementary Fig. 13.

5. Fig. 4d, e - Do the three glial types have distinct anatomical distributions or morphology? Since two glial types map to mouse astrocytes that are present in the telencephalon or not, it would be important to know if these types show a similar segregation in the octopus nervous system that might point to shared functions or homologous structures.

We agree with the reviewer that this is valuable information that would add to the strength of the manuscript. In order to verify the anatomical localization of the distinct glial types we performed additional multiplexed hybridization chain reaction experiments. We designed probe sets for genes that were specifically expressed in GLIA1 (*gat1*), GLIA2 (*hhex, wsc*), and GLIA3 (*cnfn, pkl*) and combined these with the pan-glial marker *apolpp*. No signal was observed for genes specifically expressed in GLIA2 and GLIA3 for either probe sets despite multiple trials, higher probe and amplifier concentrations. GLIA2 (104 cells) and GLIA3 (25 cells) subtypes are relatively small populations compared to GLIA1 (537 cells) and they do not have a lot of highly expressed marker genes. However, our marker for GLIA1 (*gat1*) nicely labeled the GLIA1 population together with GABAergic neurons. By using the pan-glial marker *apolpp* we could then conclude that double positive cells for *gat1* and *apolpp* are GLIA1, *gat1+ apolpp-* cells are GABAergic neurons and that *apolpp+* but *gat1-* cells likely represent cells belonging to the GLIA2/3 subtypes. The HCR together with marker gene expression allowed us to put a new hypothesis forward, that GLIA1, located in the neuropil throughout the brain, are the functional equivalent of astrocytes, while GLIA2 are a glial subtype dedicated to the OL plexiform layer and GLIA3 might represent astrocytes lining the brain, and perhaps taking part in the hemolymph-brain barrier. We included these results in the supplementary materials as Supplementary Fig. 14-15 and describe it in line 527-540.

6. Fig. 4a - It would be helpful to include a visual summary of the comparisons of developmental origins that may be contributing cell type homologies, e.g. mouse microglia and octopus hemocytes.

The resulting homologies that were less significant (<0.25) are listed in Supplementary Data 4. We do not have a good idea on how to make a visual summary that would combine all this information. We have tried to make visual summaries of expression patterns (Supplementary Fig. 13,16-17) and a phylogenetic cell taxonomy (Fig. 6a), but do not fully understand what type of representation the reviewer is aiming at. We hope the additional data in the supplemental figures gives sufficient information on genes driving cell type homologies.

7. Fig. 6b - Do any of the TFs shown in the heatmap have known roles in octopus development or are related to the development of homologous types in other species? Does TF expression drive cell type homologies across species or is this due to other classes of genes? The comparative analysis should be explained in more detail to show which functional classes of genes support the cell type similarities and what are the closest matches in fly and mouse for octopus cell types with no reported match. Are all of these novel types or are they related to types with more clear cross-species matches?

a) Do any of the TFs shown in the heatmap have known roles in octopus development or are related to the development of homologous types in other species?

Initially, we used TF merely as markers for cell types. Triggered by this reviewer's comment, we did a more profound literature search on these transcription factors in cephalopods and other species. We

found that indeed, many of these TF label similar areas in other cephalopods or have conserved roles in other species (also see above, lines 665-686).

b) Does TF expression drive cell type homologies across species or is this due to other classes of genes? The comparative analysis should be explained in more detail to show which functional classes of genes support the cell type similarities and what are the closest matches in fly and mouse for octopus cell types with no reported match. Are all of these novel types or are they related to types with more clear cross-species matches?

We investigated which genes underlie the putative cell-type homologies that came out of the SAmapping analysis. These were both TF as well as other genes and prepared two additional Supplementary Fig. 16-17 (lines 559-563 and lines 590-594). Although we found conservation of function for many TF genes, which suggests TF could drive cell type homologies, we cannot finally answer the reviewer's question at this time. Of note, during this analysis we found the following error; LOC115212290 which was mistakenly annotated as *camkII* (calcium/calmodulin-dependent protein kinase) was in fact cAMP-dependent protein kinase type II regulatory subunit or pka-R2. We thank the reviewer for this question and we revised the gene name throughout the manuscript.

8. Fig. 7b,c - How similar are the transcripts for different protocadherin genes? How did you establish that the probes for *pcdh15* and *pcdh24* label only those transcripts?

Since the protocadherin gene family expanded recently in octopus, these genes are indeed fairly similar (mean similarity is 60% on the nucleotide level). But contrary to the vertebrate clustered PCDH, they do not have shared exons. In order to understand their phylogenetic relationships, we performed a phylogenetic analysis on all octopus protocadherins. We aligned nucleotide sequences by MAFFT L-INS-I (v7.037b) and performed 1000 iterations using the Smith-waterman algorithm. Gaps were removed by trimAL (v1.2. rev59). A Bayesian method was used to reconstruct the phylogenetic trees (MrBayes v3.2.6, ngen=6000000, nchains=22). If we construct a percent similarity matrix for *pcdh24* together with the *pcdhs* that are most similar to *pcdh24*, we find that they share 50-51% similarity on the nucleotide level. *Pcdh15* shared 55% of its identity with its closest neighbor.

We designed our colorimetric *in situ* probes to minimize aspecific binding. Probes for *Pcdh15* (677 bp) and *pcdh24* (909 bp) are both of sufficient length to ensure binding to their specific targets. Moreover, we have generated sense probes that do not show any staining. For *pcdh15* and *pcdh24* we also performed multiple replicates and signal that was not consistent was considered as background. In addition, we blasted our probe sequences to the whole octopus genome and found that *pcdh15* is very distinct from other protocadherins (no hits) while *pcdh24* shares more sequence similarity to other genes (first hit shares 39% query cover, 68.13% sequence identity). To avoid confusion with vertebrate PCDHs (also comment reviewer #1), we renamed *pcdh15* to *pcdhO1* and *pcdh24* to *pcdhO2*.

We thank the reviewer for this comment and to be more transparent we added the following sentence to our methods section: "*Probes were designed to be between 500-1000 bp in length and were blasted against the genome to ensure specificity.*"

9. All of the ISH images would benefit from a corresponding schematic like Supplementary Fig. 10 that shows the distribution of cells that are positive for the marker. It is difficult to see the ISH stain in many of the figures.

We extended the supplement with a schematic illustrating representative expression patterns of all *in situ* hybridizations that were shown in the manuscript (Supplementary Fig. 23). We do think that Supplementary Fig. 10 still provided additional information regarding the co-localization and organization of specific cell-type markers and is now called Supplementary Fig. 24.

Reviewer #3 (Remarks to the Author):

Styfahls et al manuscript presents a novel single cell dataset of the developing octopus brain. I found the study to be of interest and a valuable contribution to the community for the study of cell type evolution across species. This dataset can also be beneficial to further improve cross species analysis methods, which is still an under researched area that can shed light on human biology of disease. I commend the authors for the ingenious approach used for the genome annotation. The web application hosting the dataset is also a useful tool that will help reusing the data. I'm of the opinion that this work is a good fit for Nature Communications. However, before it can be published, I would like the authors to address a couple of comments which will improve the overall analysis quality and strengthen the manuscript as a whole.

Major points

1. The authors cite reference 6 as “extensive characterization of the adult octopus brain”. Can the authors provide a comparison between the classical literature reference and the data presented in the manuscript? How many known cell types were found in the single cell data?

This is indeed a valid, but very difficult question to answer. Young described the anatomy of the adult nervous system of *Octopus vulgaris* mostly based on histological observations and Golgi stains¹⁵. This resource also includes a description of the cells present in the different lobes of the octopus brain¹⁶. Although on occasion, the morphology of particular cell types is mentioned (e.g. amacrine cells, bipolar, unipolar cells), a comprehensive overview of all distinct morphological cell types is missing. However, we can use the nuclear size as a metric for the number of cellular phenotypes¹⁶. Young classified the cells based on their location and created 5 bins for nuclear size (<5 um, 5-10 um, 10-15 um, 15-20 um, 20-25 um). A total of **116 different cellular phenotypes** (number of lobes x number of bins observed in each lobe) can be distinguished when making the following assumptions: 1) the chosen bins are accurate to discriminate cell types and 2) different brain territories or lobes possess different cell types. This number can therefore be an overestimation, since we show here that several cell types are distributed in different brain regions (e.g. serotonergic cells) or an underestimation since similar nuclear size does not necessarily result in a single cell type. We do not presume to have identified all the cell types present in the octopus brain, therefore a comparison between our data (42 cell types) and the literature (116 cellular phenotypes) remains difficult. Moreover, considering that this post-embryonic brain consists only of 200.000 cells while the adult nervous system is made up of 200 million cells, it is highly likely that not all cell types described in the adult nervous system are present. A recent preprint for the squid optic lobe⁷ showed that the cell complement differs significantly between hatchlings and adults, which suggests that maturation is still ongoing. To what extent these octopus hatchling brain cell types can be compared with adult brain cells is future work.

Since there was not much a priori knowledge relating to cell type diversity within the larval octopus brain, we had no reference list of 'expected cell types'. Only a few expression studies regarding **neurotransmitter and neuropeptide expression** in developing cephalopod brains were published. We added this to the introduction after our literature review (lines 76-79). We used these data as well to sum up the most expected findings (lines 339-345 and 353-355).

Aside from functionally different neuronal subtypes, expected cell types **based on morphology** and/or **spatial localization** were also identified. For example, we could annotate the amacrine cells that make up the vertical lobe. In addition, we could also identify a cell type that is specific to the subesophageal mass; this is significant since in the posterior lateral pedal lobe (in the SUB) in the adult brain contains the

so-called 'giant cells'¹⁵. These cells can be distinguished early in development and while they are giant in decapods, these have a conspicuous pear-shape in octopods. We thus propose the SUB might represent these pear-shaped homologs of the giant cells. We also identified different cell types within the outer and inner granular layer of the optic lobe, which is in line with the Golgi-stain based description of cells within the laminated cortex by Young. Although these might represent more numerous cell types, an alternative hypothesis is that the cells in the optic lobe are more mature at this developmental time point. Wherever possible, we added this information to the manuscript.

2. How many are missing? Did the authors undersample and/or oversample some known functional groups?

Conversely, it was even less trivial to highlight 'unexpected' or 'missing' cell types. However, we summarize our main findings here (and throughout the manuscript).

a) A limited amount of precursor cells: Although the octopus paralarva is free-swimming at this developmental life stage and shows interesting innate behaviors such as predation and positive phototaxis, we presumed that there would be clear populations of progenitor cells connected to immature neurons and more mature neurons in this 'early life stage'. An unexpected finding was that we identified only a small population of precursor cells which we believe are the remnants of the lateral lip, the transient embryonic neurogenic niche in octopus, and no clear sign of neurogenesis or a neurogenic population. We believe this suggests that the recently hatched paralarvae are transitioning from an embryonic growth phase to paralarval life and are not generating new neurons at this stage.

b) A large diversity of mature optic lobe cell types: We looked at the spatial localization of several cell types within the octopus brain and for the most part we identified optic lobe cell types and based on our iCytoTRACE analysis these cell types were often considered to be more mature. This suggests that the optic lobe is more mature compared to other brain regions at this point and already exhibits a surprising amount of diversity (lines 455-457).

c) Many rare peptidergic neurons: The central constellation was an unexpected finding. Whether this is a trait of centralized invertebrate brains (*Drosophila*, *Octopus*), we do not know. We hypothesized above that the central cluster likely represents rare cell types that are likely peptidergic. Of note, we also identified two cell types (CCAP and PEP-burs) that express peptides involved in ecdysis in arthropods. We added this to the manuscript in lines 412-414.

To make it clear that more cell types are likely present, we added lines 755-757 to the conclusion.

As mentioned above, we could not rely on any molecular information regarding cell types in the hatchling octopus brain as only tissue-wide expression data or adult cell type morphology was known, and comparisons with the adult octopus nervous system remains difficult. Our study aims to make a start in identifying the cell types at the molecular level. We can estimate the number of cells we expect to find (see above) but this is also merely an estimation. It is therefore impossible to say at this stage how many more we might find when we sample deeper, and how many we missed here. Since we do not presume that we have captured all the cell types within this developing brain, the real number of cell types is still obscure. Our analysis likely revealed the most prevalent cell types that have been sampled sufficiently, while the central constellation contains rare cell types.

3. 10x data processing:

a) Why do the authors have a different number of genes for nuclei and whole cell? Did the authors use a different reference genome?

Indeed, we find that the total number of genes detected in the nuclei sample is higher than for the cells although the genome reference used was identical. The percentage of reads mapping to intronic regions is higher in nuclei (26.60% in nuclei vs 19.30% in cells), since mRNA originating from the nuclei also

contains pre-mRNA. In order to count the maximum number of genes, we also counted the reads that mapped to intronic regions as is recommended when analyzing single nuclei RNA-seq data (--include-introns option in Cell Ranger). We counted the intronic reads in both the nuclei and cells in order to compare and integrate our results. This resulted for both samples in better mapping statistics. In newer versions of Cell Ranger, this is now the default option also for single cell RNA-seq data. We also found that more lncRNAs were detected within the nuclei sample versus the cells, and it is likely that this is an additional factor that increases the gene counts in the nuclei.

b) Can the authors justify why they filter cells with both too high or too low gene counts and with more than 5% mito reads? Did the authors find any evidence between these variables and the presence of technical artifacts such as doublets or ambient RNA?

We optimized our single-cell suspension protocol in order to obtain highly viable cells and a low % of doublets. Based on the LUNA-counter results 97.4% of all cells were singlets, 2.3% were doublets and 0.2% were triplets. Since we knew that a small % of cells were doublets we wanted to be as stringent as possible and only keep high-quality cells and nuclei within our dataset. Therefore, we filtered out cells and nuclei with high gene counts.

In order to exclude low-quality cells or empty droplets with ambient RNA we also filtered out cells with low gene counts. For the nuclei this threshold was put at 400, while for the cells this was 800. The rationale behind this higher threshold was that we could observe an abnormally high population of cells with low gene counts (see figure below). This population had low gene counts and a low number of reads. We analyzed this population in depth before deciding to exclude it from our subsequent analysis. We contributed this to a technical artefact from the single cells, since this population was absent in the nuclei.

Fig. R2. Abnormal distribution of number of genes per cell in the cells.

Lastly, we removed all cells with a high percentage of mitochondrial RNA since this could point towards low-quality and/or dying cells. We added the following to the manuscript to explain our rationale (lines 216-218): Nuclei and cells with too high (>4000) or too low (<400 for nuclei, <800 for cells) gene counts were filtered out “to exclude doublets and empty droplets.” Cells with a higher percentage (>5) of mitochondrial RNA were regressed out “since these are likely of low-quality and possibly dying.”

c) Following up from point 2b, the authors were not able to annotate the center of tSNE plots in Fig1 and subsequent figures. Did the authors look in detail at those cells and found any evidence of low quality metrics? It is not clear from the supplementary figure if a deep analysis was done and the rationale for keeping the non-annotated cells is also not discussed. Did the authors try to change the filtering parameters and observe any changes in the tSNE graph? Could a different embedding have helped finding whether the central region of the tSNE is mostly low quality transcriptomes or perhaps enrich the cell type annotations?

All reviewers had similar questions regarding the central constellation. We therefore further studied this enigmatic group of cell clusters in more detail. First, we compared our data with another invertebrate

brain (*Drosophila*), which resulted in Fig. R1. In addition, we investigated their quality, whether they could be subclustered and whether there were common features so we could hypothesize potential functions.

Table 1: Ideal clustering parameters for the different samples.

SAMPLE	PC	RES	K	TOTAL CLUSTER NUMBER
CELLS_MOST_STABLE	150	1.4	10	72
CELLS_MOST_PERC	150	0.6	16	35
NUCLEI_MOST_STABLE	50	3	10	67
NUCLEI_MOST_PERC	150	0.6	30	21
INT_MOST_STABLE	150	2	10	87
INT_MOST_PERC	50	0.6	30	27

a) *Are these cells of low quality?*

To investigate what is the biological difference between ‘central’ and ‘peripheral’ cells we grouped the cells based on their localization on the tSNE plot and investigated standard quality metrics, immediate early gene expression and transcription factors that were differentially expressed (Supplementary Fig. 5). The central constellation consists of high-quality cells (also see Supplementary Fig. 2d). This cluster has a high number of features, high counts and low mitochondrial gene expression (Supplementary Fig. 5b). The central constellation is clearly divided in two based on the expression of *vglut* and *vacht* and therefore consists mainly of glutamatergic and cholinergic neurons (Fig. R1a, 2b). When we compare our dataset with another invertebrate brain atlas (fly) we observe a striking similarity (Fig. R1b). We added our findings to the manuscript (lines 356-365)

b) *Are these immature cells?*

We performed an additional analysis (iCytoTrace), which is based on the hypothesis that gene counts generally decrease with differentiation². The output of this analysis is visualized on the t-SNE plot in Supplementary Fig. 6. We found that the cells in the central constellation are predicted to be less differentiated. However, since there are peripheral clusters (SUB, SERT, IGL3) that also show high transcriptional diversity (i.e. less differentiated) we do not think that immaturity is the biological reason setting the central constellation apart. We also think it is unlikely that immature neurons already highly express markers of functional neurons such as *vacht* and *vglut*. We added our methods (lines 243-245) and findings (lines 365-370) to the revised manuscript.

c) *Do these cells represent rare cell types?*

When comparing our dataset with another published invertebrate brain dataset; the fly brain atlas, a similar central constellation was observed (Fig. R1a, b). In the fly brain, doubling the number of sequenced cells from 57k to 118k^{3,4}, decreased the size of the central constellation and increased the number of distinct and separated clusters. The total number of cells present in the adult fly brain was estimated to be 199,380 ± 3,400 cells⁵. Even after sequencing around 60% of all cells, the central constellation was not resolved (Fig. R1b, d). Intriguingly, they did find that the central constellation mostly represents cell types that are localized within the central brain, while the more divergent clusters are cell types present in the optic lobe or very distinct cell types such as glial cells (Fig R1d)⁴. We also observed that the majority of stable (non-central-constellation) clusters we mapped spatially originated from the optic lobes. On the other hand, it is known that there are cell types present in very small numbers within the fly brain so this could also reflect the abundance of rare cell types, while in the optic lobe there might be more repetition and re-use of the same cell types. Whether the central constellation indeed represents more central brain cell types in octopus we could not distinguish at this stage. The issue was that ‘marker’ genes of the central constellation were not unique, but were often also expressed in the surrounding cell types. Therefore, we were unable to localize these cells within the brain via *in situ* hybridization. To circumvent this issue, we performed an additional analysis to identify whether the central constellation

was more central brain-like or optic lobe-like by using publicly available bulk RNA-seq data of different regions of the adult octopus brain (Supplementary Fig. 7). We grouped the supra-esophageal mass and the sub-esophageal mass together as central brain and identified differentially expressed genes between the central brain and the optic lobes. We then used the module scoring function in Seurat to score all the cells based on the top 100 optic lobe genes and the top 100 central brain genes. We mostly identified cell types as 'optic lobe' that were already spatially mapped there (OGL2-DOP, IGL2-GLUT/DOP, IGL1-OA) and cell types we expected to map to the central brain (SERT, SUB, VL, GLUT4) indeed also mapped to their predicted location. Intriguingly, also populations whose locations are still unknown such as DOP1, DOP2, DOP3, OA showed high expression levels of adult optic lobe specific genes. Nevertheless, we found that the central constellation showed both expression of central brain and optic lobe specific genes. We therefore concluded that the central constellation had no specific spatial localization, but was a mixture of different rare cell types. Moreover, the presence of such big central constellations was also observed in recent single-cell studies on squid embryonic head⁶ and adult brain tissue⁷. We added our methods (lines 245-247) and findings (lines 384-391) to the revised manuscript.

d) Is there heterogeneity within this cluster?

To further understand what cell types might be hidden in the central constellation, we created a subset of the data containing only the central constellation and identified the most highly variable genes for these sub clusters. Still, even after subclustering, most of the data resided in a central constellation (Supplementary Fig. 8). We found that within this subset neuropeptides are overrepresented in the highly variable genes, and are especially marking the small clusters, suggesting that the central constellation consists out of rare cell types that are neuropeptidergic (Supplementary Fig. 8c). We also identified transcription factors that are highly variable, including *rn/sqz*, a TF that regulates neuropeptidergic cellular identity (Supplementary Fig. 8d)⁸. Together, these analyses suggest that neuropeptidergic expression by rare cell types might be important in this young stage cephalopod brain. We added our methods (lines 247-249) and findings (lines 391-396) to the revised manuscript.

Fig. R1. Central constellation in *Octopus vulgaris*. **a** The central constellation consists mainly of glutamatergic (*vglut*) and cholinergic neurons (*vacht*). **b** A similar central constellation can be observed in the aging fly brain³. **c** Most of the neuronal cell types that surround the central constellation are localized in the optic lobes. **d** After sequencing 60% of all cells present in the fly brain, the central constellation remains and represents mainly central brain cell types.

d) Data integration: the authors do not discuss the integration of whole cell and nuclei. The authors must at the very least discuss the advantages and limitations of each technology and how each and both combined empowered the analysis that are presented in the manuscript. I strongly disagree that “by using a dual approach in sequencing both cells and nuclei [they] have overcome any technical bias associated with each sequencing technology”. There is a considerable amount of literature discussing the comparison of the two technologies. Using both and ignoring the artifacts that each carries and that consequently will be present and likely affect the downstream analysis cannot be dismissed with such a sentence.

All reviewers raised a similar question. We therefore prepared an additional figure (Supplementary Fig. 3) to highlight the differences between the two datasets. Although most cell types have cells originating from both samples, we show in Supplementary Fig. 2c that the proportion of cells per cell type originating from cells versus nuclei is not always equal. Out of the 42 stable clusters, we observed 24 clusters with a slightly higher proportion of cells versus nuclei. We found that the clustering and the number of stable clusters was similar for both samples. We did see that through data integration we were able to distinguish more cell types and that obtained stable clusters were larger. We therefore decided to continue the annotation on the integrated dataset. We investigated the effect of integrating the cells and the nuclei by looking at the top 10 differentially expressed genes between the two datasets (Supplementary Fig. 3d). We found that in the nuclear RNA seq more lncRNA's were upregulated. In the cellular RNA seq we identified heat shock proteins and genes that might be linked to a stress response such as immediate early genes. We added these findings to the revised manuscript (lines 310-320).

Minor points

- 1. Line 102, sentence starting with “Since the development of single-cell RNA sequencing technologies...” does not contribute to neither the previous nor the following sentence. Please rewrite.**

We removed this sentence from the manuscript.

- 2. Fig.3 consider changing the order of the panels by renumbering c->b, e->c, g->d, b->e, d->f and f->g
Improve the readability by clearly indicating in the figure in the right corresponds to the magnification of the rectangle.**

We adjusted the labels of Fig. 3 as suggested. In addition, we indicated the correspondence of the magnification with the overview images with grey lines.

- 3. Fig.5 the staining of tmtc4 is not very clear, and in tSNE the expression of the gene is fairly scattered
Panel d - what do the different colors above the heatmap represent?**

Fig. 5c. We agree that the expression for this gene is low. We selected this marker gene to ensure specific expression only in the vertical lobe (compared to *pka-R2* which also has lower expression in different clusters). On the t-SNE plot you can also observe that not all cells express this gene, which explains the sparse expression in the vertical lobe. We adjusted the color scaling for the t-SNE plot to highlight the high specific expression within the vertical lobe in Fig. 5c.

Fig. 5d. The colors represent the different cell types. We agree with the reviewer that this additional delineation of the different cell types is confounding. We removed the labels on Fig. 5d and also on Fig. 4b.

4. Can the authors elaborate about the possible orthologue for mki67 described in the non-neuronal cell types section?

We discuss a possible orthologue for MKI67, a common vertebrate cell proliferation marker. Blastp searches against all other species also showed similarities with other proteins (mucin, proteoglycan,...), hence this analysis alone does not prove it is a true MKI67 orthologue. However, hits for MKI67 include a variety of species such as *Xenopus*, *Mytilus*, *Mizuhopecten*, *Branchiostoma* and *Anneissia*. In addition, this MKI67 gene (LOC115228146) is highly expressed within the identified precursor population. To verify whether this is in fact an MKI67 gene we performed additional analyses with alphaphold db and interproscan and found protein domains indicating that in fact this is a MKI67 orthologue (FHA, PP1-b, antigen KI-67 like protein domain). We therefore removed “a possible orthologue” from the manuscript to avoid any confusion.

5. Fig.6 - the authors should include the full results of the analysis summarized in Fig.6a as a supplementary table

We thank the reviewer for this comment and have composed these results in Supplementary Data 7.

6. Discussion - as the authors well document in introduction, cell type diversity in a mollusk has already been reported. Please rewrite the first sentence of the discussion section.

Indeed, while this paper was in revision, three more preprints describing cephalopod single cell data have come out. We have deleted these sentences. We thank the reviewer (also a comment from reviewer #1) for this comment.

7. Fig.S3 - similar to minor point 3, the expression of GABA does not look specific

We agree that the expression of both *gad* (LOC115220799) and *vpat* (LOC115215891) on the t-SNE representation looks fairly scattered and divided in two sub-clusters. However, on the UMAP-embedding these two populations merge into one. In addition, the co-localization of both *gad* and *vpat* provides additional support that the GABA cluster are indeed GABAergic neurons. We show this co-expression and different embeddings here for the reviewer in Fig. R3. There is indeed low expression outside of the GABA cluster and to make it more clear for the readers we have adjusted the color scaling to illustrate that expression is highest within the GABA cluster in Supplementary Fig. 3.

Fig. R3. Co-expression of Glutamate decarboxylase (*gad*) and vesicular GABA transporter (*vgat*) is visualized on the t-SNE and UMAP-embeddings.

8. Fig.S9 - this figure is extremely hard to read, consider adding this to the web portal instead of having it as a supplementary figure

We compiled an additional supplementary data file with the gene id lists for the investigated gene families: ZnF, GPCR and PCDH. We added the following sentence in the methods section: “*Gene lists are available as Supplementary Data 8*”.

References

1. Stuart, T. *et al.* Comprehensive Integration of Single-Cell Data. *Cell* **177**, 1888–1902 (2019).
2. Gulati, G. S. *et al.* Single-cell transcriptional diversity is a hallmark of developmental potential. *Science* (80-.). **367**, 405–411 (2020).
3. Davie, K. *et al.* A Single-Cell Transcriptome Atlas of the Aging *Drosophila* Brain. *Cell* **174**, 982-998.e20 (2018).
4. Janssens, J. *et al.* Decoding gene regulation in the fly brain. *Nature* **601**, 630–636 (2022).
5. Raji, J. I. & Potter, C. J. The number of neurons in *Drosophila* and mosquito brains. *PLoS One* **16**, 1–11 (2021).
6. Duruz, J. *et al.* Molecular characterization of cell types in the squid *Loligo vulgaris*. *bioRxiv* 2022.03.28.485983 (2022).
7. Gavriouchkina, D. *et al.* A single-cell atlas of bobtail squid visual and nervous system highlights molecular principles of convergent evolution. (2022).
8. Allan, D. W., St. Pierre, S. E., Miguel-Aliaga, I. & Thor, S. Specification of neuropeptide cell identity by the integration of retrograde BMP signaling and a combinatorial transcription factor code. *Cell* **113**, 73–86 (2003).
9. Gleadall, I. G. *Octopus sinensis* d'Orbigny, 1841 (Cephalopoda: Octopodidae): Valid species name for the commercially valuable East Asian common octopus. *Species Divers.* **21**, 31–42 (2016).
10. Zarrella, I. *et al.* The survey and reference assisted assembly of the *Octopus vulgaris* genome. *Sci. Data* 1–8 (2019). doi:10.1038/s41597-019-0017-6
11. Ren, Q., Awasaki, T., Wang, Y. C., Huang, Y. F. & Lee, T. Lineage-guided Notch-dependent gliogenesis by *Drosophila* multi-potent progenitors. *Dev.* **145**, (2018).
12. Castellanos-Martínez, S., Arteta, D., Catarino, S. & Gestal, C. De novo transcriptome sequencing of the octopus *vulgaris* hemocytes using illumina RNA-Seq technology: Response to the infection by the gastrointestinal parasite *Aggregata octopiana*. *PLoS One* **9**, (2014).
13. Collins, A. J., Schleicher, T. R., Rader, B. A. & Nyholm, S. V. Understanding the role of host hemocytes in a squid/*Vibrio* symbiosis using transcriptomics and proteomics. *Front. Immunol.* **3**, (2012).
14. Yoshida, M. A., Shigeno, S., Tsuneki, K. & Furuya, H. Squid vascular endothelial growth factor receptor: A shared molecular signature in the convergent evolution of closed circulatory systems. *Evol. Dev.* **12**, 25–33 (2010).
15. Young, J. Z. *The anatomy of the nervous system of Octopus vulgaris*. (Oxford University Press, 1971).
16. Young, J. Z. The number and sizes of nerve cells in *Octopus*. *Proc. Zool. Soc. London* **140**, 229–254 (1963).
17. Styfhals, R., Seuntjens, E., Simakov, O., Sanges, R. & Fiorito, G. In silico Identification and Expression of Protocadherin Gene Family in *Octopus vulgaris*. *Front. Physiol.* **9**, 1–8 (2019).
18. Albertin, C. B. *et al.* Genome and transcriptome mechanisms driving cephalopod evolution. *Nat. Commun.* (2022). doi:10.1038/s41467-022-29748-w
19. Rubinstein, R., Goodman, K. M., Maniatis, T., Shapiro, L. & Honig, B. Structural origins of clustered protocadherin-mediated neuronal barcoding. *Semin. Cell Dev. Biol.* **69**, 140–150 (2017).
20. Grau-Bové, X. & Sebé-Pedrós, A. Orthology Clusters from Gene Trees with Possvm. *Mol. Biol. Evol.* **38**, 5204–5208 (2021).

REVIEWER COMMENTS

Reviewer #1 (Remarks to the Author):

I thank the authors for their efforts to address concerns raised by myself and the other referees. In response to these concerns, the authors have undertaken heroic efforts to understand the bounds of their data, only to essentially return the same answers. Based on these further analyses, it seems that the limiting factor in resolving these cell types are the data. That is, it seems that an analysis of roughly 17,000 cells out of an estimated 200,000 cells in the embryonic brain of an animal with most elaborate invertebrate nervous system can resolve the top 42 out of an estimated 100-150 cell types based on classic studies - accounting for less than half of the cells reported (my back of the napkin calculation puts it at ~44% - $\sim 7600/17000$). This seems quite low, even for an emerging model system. In designing this study, how did the authors settle on this number of cells? Would further sequencing allow the authors to identify some of the "rare" cell types?

At minimum, the manuscript would benefit from pointing out the discrepancy between the number of anticipated cell types and those that they were able to resolve – the authors do a thorough job in their response to referees to highlight the difficulty of the problem – and clarifying this in the manuscript itself will help readers to understand the data presented. I acknowledge that the authors nod to this – indicating that they have sequenced ~9% of the total number of cells, and that “42 clusters is likely an underestimation of the total number of cell types”. However, I think the historical context strengthens the the suggestion/explanation that the central constellation is likely “rare cell types” and provides important context to the reader.

Minor points:

In the text the authors state that less than 60% of the scRNAseq reads mapped, but in the authors response, they indicate mapping was 80% or above. Why the discrepancy?

I am confused about Sup fig 17 – why are the octopus cells labeled with fly names and vice versa – are these simply typos?

Reviewer #2 (Remarks to the Author):

The authors have address my comments and I support publication.

Reviewer #3 (Remarks to the Author):

Styfals et al addressed all my questions during the review processed. The present manuscript is a much improved version and I find it to be suitable for publication.

Cell type diversity in a developing octopus brain

Reply to the reviewers.

Reviewer #1 (Remarks to the Author):

I thank the authors for their efforts to address concerns raised by myself and the other referees. In response to these concerns, the authors have undertaken heroic efforts to understand the bounds of their data, only to essentially return the same answers. Based on these further analyses, it seems that the limiting factor in resolving these cell types are the data. That is, it seems that an analysis of roughly 17,000 cells out of an estimated 200,000 cells in the embryonic brain of an animal with most elaborate invertebrate nervous system can resolve the top 42 out of an estimated 100-150 cell types based on classic studies - accounting for less than half of the cells reported (my back of the napkin calculation puts it at ~44% - $\sim 7600/17000$). This seems quite low, even for an emerging model system. In designing this study, how did the authors settle on this number of cells? Would further sequencing allow the authors to identify some of the "rare" cell types?

At minimum, the manuscript would benefit from pointing out the discrepancy between the number of anticipated cell types and those that they were able to resolve – the authors do a thorough job in their response to referees to highlight the difficulty of the problem – and clarifying this in the manuscript itself will help readers to understand the data presented. I acknowledge that the authors nod to this – indicating that they have sequenced ~9% of the total number of cells, and that “42 clusters is likely an underestimation of the total number of cell types”. However, I think the historical context strengthens the the suggestion/explanation that the central constellation is likely “rare cell types” and provides important context to the reader.

We would like to point out that in established model organisms such as *Drosophila melanogaster* the % of high quality sequenced cells was 28% in the initial study, i.e. 57,000 out of 200,000 cells (Davie et al., 2018). However, even sequencing 60% of all cells in a follow-up study did not resolve all cell types (Janssens et al., 2022). Moreover, a preprint reported similar numbers of cells within the squid whole head including the 8 arms, which comprises a lot more cells (Duruz et al., 2022). Hence, we do think the number of cells and the percentage analyzed is not uncommon for an initial work.

We do agree that more cells will likely resolve some additional cell types, but only a massive approach might be able to resolve all. A recent preprint of the adult human brain also found a similar unresolved population of cells which were called ‘splatter neurons’, and identified these as highly complex and as likely undersampled (Siletti et al., 2022). This group of highly heterogeneous splatter neurons uniquely comprised both glutamatergic and GABAergic neurons that did not separate after clustering and expressed neurotransmitters and neuropeptides in a complex combinatorial pattern. Per definition, clustering will try and group cells, and if indeed these central clusters represent rare cell types (i.e. one cell in each hemisphere) it is unlikely these cells will cluster separately until all the cells in the brain are sequenced, and even then this might not be the case.

We therefore agree with the reviewer that highlighting the shortcomings of our study would be beneficial for the field and therefore added the following text to the manuscript (lines 757-766):

“Based on the work by J.Z. Young, the number of real cell types likely ranges between 100-150 in the adult octopus nervous system (Young, 1963, 1971). By using the difference in nuclear size and anatomical location inferred from classical Golgi stainings as a metric, we can distinguish 116 different cellular phenotypes. In this study, we were able to identify only 42 cell types in the hatchling brain, out of the putative 116 cell types present in the adult. Since this developing octopus brain still needs to grow, the number of cell types will also likely increase with age. Moreover, the large number of cells in the central constellation likely comprises many rare cell types. Increasing the number of sequenced cells might shed light on the heterogeneity of the central constellation and will likely resolve more cell types in future studies.”

Minor points:

In the text the authors state that less than 60% of the scRNAseq reads mapped, but in the authors response, they indicate mapping was 80% or above. Why the discrepancy?

We assume the reviewer refers to these percentages mentioned within the text;

“With this new annotation, the percentage of reads that mapped confidently to the transcriptome increased significantly (from 32.5% to 45.6% for the nuclei and from 49.4% to 58.8% for the cells; Supplementary Data 2).”

Within our response to the reviewers we report the % of reads mapped to the genome for the cells and nuclei, which is indeed 88.4% and 80%, respectively. The discrepancy therefore arises from reporting the statistics for the genome versus the transcriptome. A detailed overview of the mapping statistics can be found in Supplementary Data 2. For clarity, we also indicate the mapping % against the genome within the manuscript (lines 296-297):

“We were able to map 80-88.4% of all reads to this genome, for the nuclei and cells, respectively.”

I am confused about Sup fig 17 – why are the octopus cells labeled with fly names and vice versa – are these simply typos?

Supplementary Figures 16 and 17 visualize the expression levels of the genes that underly the putative cell type homologies between octopus-mouse (fig16) and octopus-fly (fig17). Hence, panel a in fig 17 shows the expression of these genes in octopus and panel b in the fly brain (indicated by the species icons). The annotation in the sidebar on the left represents the putative homologous cell type in the other species. We agree with the reviewer that this might be confusing and therefore we added the following sentence to the legends of Fig 16 and 17:

“Color code and cell type annotation in the left sidebar indicate the putative homologous cell types in mouse (a) and octopus (b).”

“Color code and cell type annotation in the left sidebar indicate the putative homologous cell types in fly (a) and octopus (b).”

References

- Davie, K., Janssens, J., Koldere, D., De Waegeneer, M., Pech, U., Kreft, Ł., ... Aerts, S. (2018). A Single-Cell Transcriptome Atlas of the Aging *Drosophila* Brain. *Cell*, 174(4), 982-998.e20. <https://doi.org/10.1016/j.cell.2018.05.057>
- Duruz, J., Sprecher, M., Kaldun, J., Alsoudy, A., Tschanz-Lischer, H., van Geest, G., ... Sprecher, S. G. (2022). Molecular characterization of cell types in the squid *Loligo vulgaris*. *BioRxiv*, 2022.03.28.485983. Retrieved from <http://biorxiv.org/content/early/2022/03/29/2022.03.28.485983.abstract>
- Janssens, J., Aibar, S., Taskiran, I. I., Ismail, J. N., Gomez, A. E., Aughey, G., ... Aerts, S. (2022). Decoding gene regulation in the fly brain. *Nature*, 601(7894), 630–636. <https://doi.org/10.1038/s41586-021-04262-z>
- Siletti, K., Hodge, R., Albiach, A. M., Hu, L., Lee, K. W., Lönnerberg, P., ... Linnarsson, S. (2022). Transcriptomic diversity of cell types across the adult human brain.
- Young, J. Z. (1963). The number and sizes of nerve cells in *Octopus*. *Proceedings of the Zoological Society of London*, 140(2), 229–254. <https://doi.org/10.1111/j.1469-7998.1963.tb01862.x>
- Young, J. Z. (1971). *The anatomy of the nervous system of Octopus vulgaris*. London, UK: Oxford University Press.

Reviewer #2 (Remarks to the Author):

The authors have address my comments and I support publication.

Reviewer #3 (Remarks to the Author):

Styfals et al addressed all my questions during the review processed. The present manuscript is a much improved version and I find it to be suitable for publication.